# FSscore: A Machine Learning-based Synthetic Feasibility Score Leveraging Human Expertise

**Rebecca M. Neeser**[12]    **Bruno Correia**[2]    **Philippe Schwaller**[13]
[1] Laboratory of Artificial Chemical Intelligence (LIAC), EPFL, Switzerland
[2] Laboratory of Protein Design and Immunoengineering (LPDI), EPFL, Switzerland
[3] National Centre of Competence in Research (NCCR) Catalysis, EPFL, Switzerland
{rebecca.neeser,bruno.correia,philippe.schwaller}@epfl.ch

## Abstract

Determining whether a molecule can be synthesized is crucial for many aspects of chemistry and drug discovery, allowing prioritization of experimental work and ranking molecules in *de novo* design tasks. Existing scoring approaches to assess synthetic feasibility struggle to extrapolate to out-of-distribution chemical spaces or fail to discriminate based on minor differences such as chirality that might be obvious to trained chemists. This work aims to address these limitations by introducing the Focused Synthesizability score (FSscore), which learns to rank structures based on binary preferences using a graph attention network. First, a baseline trained on an extensive set of reactant-product pairs is established that subsequently is fine-tuned with expert human feedback on a chemical space of interest. Fine-tuning on focused datasets improves performance on these chemical scopes over the pre-trained model exhibiting moderate performance and generalizability. This enables distinguishing hard- from easy-to-synthesize molecules and improving the synthetic accessibility of generative model outputs. On very complex scopes with limited labels achieving satisfactory gains remains challenging. The FSscore showcases how human expert feedback can be utilized to optimize the assessment of synthetic feasibility for a variety of applications.

## 1 Introduction

The ability to assess the synthetic feasibility of a small molecule is of great importance in many different areas of chemistry, notably in early drug discovery stages. Trained chemists traditionally perform this task through intuition or retrosynthetic analysis, allowing them to decide, which molecules are likely possible to synthesize and prioritize based on synthetic complexity. However, the chemical space that might be accessible is massive, and only a small fraction has been explored. [1, 2] Furthermore, computational approaches such as virtual screening (VS) [3] in drug discovery or the recent surge of generative methods for *de novo* molecular design [4–14] emphasize the requirement for suitable tools to score synthetic feasibility quickly in an automated fashion. [15, 16].

Currently, available scores work well at discriminating feasible from unfeasible molecules in the data distribution that they were designed for but often fail to generalize. This is especially true for machine learning (ML) based scores that are unable to capture such an abstract concept as synthesizability and fail to perform well on out-of-distribution data, especially when applied in the context of generative models. [15, 17–19] However, exploring new chemical space is of great interest specifically in the context of *de novo* design or new drug modalities such as synthetic macrocycles or proteolysis targeting chimeras (PROTACs). On the other hand, synthetic feasibility cannot be merely captured by the structure as it also depends on a chemist's available resources and expertise. [20] Thus, incorporating human preference would greatly improve the practical utility of such a score. [21]

This work presents a novel approach to assess synthetic feasibility and investigates the incorporation of expert knowledge to tune the model towards a desired chemical space. We build on ideas first put forth by Coley et al. [22] when introducing the SCScore. As such, we use reaction data to pre-train our model, which implicitly informs on the difficulty of synthesizing a molecule through the relational nature of the data, while containing more information than just the number of reaction steps. Furthermore, by framing this task as a ranking problem based on pairwise preferences, we avoid the need for a ground truth score, but base the model on reported chemical reactions. [22, 21] To ultimately tune to a specific chemical space with as little data as possible we fine-tune the unbiased baseline model with expert chemist-labeled data in an active-learning type framework as inspired by MolSkill [21]. Besides the various applications as an offline scoring tool, this fully differentiable approach could also be directly used as guidance in generative models or as reward function in reinforcement learning (RL) frameworks. Compared to previous work [22–24], our representations consider stereochemistry and repeated substructures, which are crucial for determining the synthesizability of molecules. To summarize, the FSscore is trained in two stages:

1. Pre-trained on a large dataset of reactions to establish an unbiased baseline using an expressive graph neural network (GNN).
2. Fine-tuning using human feedback to focus the model towards a chemical space of interest.

Our main contributions are:

- We propose a novel approach for assessing synthetic feasibility that uses pairwise preferences and fine-tuning with human feedback to focus the model on a desired chemical space. This allows for incorporating expert knowledge and intuition.
- The method is fully differentiable, allowing it to be easily incorporated into generative models.
- Our experiments show the model can be effectively fine-tuned with relatively small amounts of human-labeled data, as little as 20-50 pairs, which is important for practicality.
- Fine-tuning improved performance on several chemical scopes, including natural products and PROTACs, demonstrating the approach's ability to adapt to new domains.

## 2   Related Work

Various methods to capture synthetic accessibility or complexity exist and can be structure-based (SA score [25], SYBA [23], GASA [26]) or reaction-based (SCScore [22], RAscore [24], CMPNN [27], RetroGNN [28], DFRscore [29]). The commonly used Synthetic Accessibility score (SA score) is rule-based and penalizes the occurrence of fragments rarely found in a reference dataset and the presence of specific structural features. [25] Thus, it captures more synthetic complexity than accessibility and fails to identify big complex molecules with mostly reasonable fragments. [15, 20] The SYBA score was trained to distinguish existing synthesizable molecules from artificial complex ones but the performance was found to be sub-optimal. [17, 23] Many of these structure-/fragment-based approaches suffer from the inability to capture small structural differences due to lacking sensitivity. Yu et al. [26] attempt to address this by using a graph representation to classify molecules into hard (HS) and easy (ES) to synthesize similarly to SYBA. The Synthetic Complexity score (SCScore) predicts complexity in terms of required reaction steps and is based on 1024-bit Morgan fingerprints. The SCScore was trained on the assumption that reactants are easier to make than products. [22] This score performs well on benchmarks approximating the length of the predicted reaction path but poorly when predicting feasibility in benchmarks using synthesis predictors. [15] The Retrosynthetic Accessibility score (RAscore) predicts synthetic feasibility with respect to a synthesis prediction tool and thus is directly dependent on the performance of the upstream model. [24] Similarly, RetroGNN classifies molecules based on retrosynthetic accessibility with the specific aim of being applied in VS. [28] Li and Chen [27] suggested a score based on a Communicative Message Passing Neural Network (CMPNN), which aims at discriminating ES from HS based on number of reaction steps using a reaction knowledge graph. Recently, the Drug-Focused Retrosynthetic score (DFRscore) was introduced by Kim et al. [29] predicting the number of reaction steps required based on a limited set of reaction templates relevant to drug discovery.

The incorporation of human feedback in ML has gained increasing attention since the introduction of RL with Human Feedback (RLHF) [30] by OpenAI, which lead to the development of popular

tools such as InstructGPT [31] and ChatGPT [32]. Learning with human feedback has also found its way into applications in chemistry: Sheridan et al. [33] trained a random forest model to predict the complexity based on human rankings. Ranking individual molecules instead of preference labeling is likely to suffer from more bias, which might be reflected in the moderate correlations of the labels to the predicted score. [34] This is avoided in the design of MolSkill, where the model learns to rank based on binary preferences made by medicinal chemists. [21] However, the scored objective, general preference, is loosely defined, and substantial pre-filtering of the training data makes the model not applicable to synthesizability and likely fails to generalize well.

## 3 Methods

### 3.1 Focused Synthesizability score

The following sections outline the method behind the novel Focused Synthesizability score (FSscore). We first train a baseline score assessing synthesizability using a graph representation and suitable message passing scheme improving expressivity over similar frameworks such as the SCScore. Secondly, we introduce our fine-tuning approach using human expert knowledge, allowing us to focus the score towards a specific chemical space of interest.

#### 3.1.1 Ranking molecules using graph embeddings

Our approach to learning a continuous score to assess the synthesizability is inspired by Choung et al. [21], who frame a similar task as a ranking problem using binary preferences. Specifically, every data point consists of two molecules for which we each predict a scalar in separate forward passes. The minimization of the binary cross entropy between the true preference and the learned score difference $\delta_{ij} := \hat{f}(m_i) - \hat{f}(m_j)$ (scaled using a sigmoid function) constitutes our training objective.

The function $f : \mathcal{M} \to \mathbb{R}$ learns to parametrize molecules as an expressive latent representation given a set of molecules $m_1, ..., m_n \in \mathcal{M}$. We represent molecules as graphs $G = (\mathcal{V}, \mathcal{E})$ with atoms as nodes $x_1, ..., x_n \in \mathcal{V}$ and bonds as edges $e_1, ..., e_m \in \mathcal{E}$ from which we can compute the line graph $L(G)$ offline iteratively to the desired depth. The transformation process to the line graph is defined such that the edges $\{e_1, ..., e_m\}$ of graph $G$ are the nodes of the line graph and these nodes are connected if the corresponding edges of $G$ share a node. The graph neural network (GNN) embedding the molecular graph consists of the graph attention network (GATv2) [35] operating on the graph $G$ and the Line Evolution (LineEvo) [36] layer operating on the line graph $L(G)$ as message passing schemes. Both GATv2 and LineEvo use an attention mechanism to update the node representations $\{h_1, ..., h_n\}$ as followed:

$$e(h_i, h_j) = a^T \sigma(W_i h_i || W_j h_j) \tag{1}$$

where $a \in \mathbb{R}^{2d'}$ and $W \in \mathbb{R}^{d' \times d}$ are learned, $\sigma$ denotes an activation function (`LeakyReLU` for GATv2 and `ELU` for LineEvo) and $||$ denotes concatenation. In GATv2, these local attention scores $e_{ij}$ are then averaged across all neighbors $\mathcal{N}$ (see Eq. 2 to obtain a normalized attention coefficient $\alpha_{ij}$ to compute the updated node representation $h_i'$ as a weighted average (see Eq. 3):

$$\alpha_{ij} = \text{softmax}_j(e(h_i, h_j)) = \frac{\exp(e(h_i, h_j))}{\sum_{j \in \mathcal{N}_i} \exp(e(h_i, h_j))} \tag{2}$$

$$h_i' = \sigma(\sum_{j \in \mathcal{N}_i} \alpha_{ij} W h_j) \tag{3}$$

with `PReLU` [37] as nonlinearity $\sigma$. In LineEvo layers, $e(h_i, h_j)$ is simply transformed using `ELU` to obtain the new node representation $h_i'$.

These transformation layers are stacked so that two GATv2 (G) layers are followed by one LineEvo (L) layer (GGLGGL). Each of these layers is followed by a readout function to obtain a molecular representation, which consists of a global max pooling layer and global weighted-add pooling as suggested by Ren et al. [36]. The intermediate molecular representations of all layers are summed and the final score $s_i = \hat{f}(m_i)$ is inferred with a multilayer perceptron (MLP). The model described above was compared to five other implementations: GATv2 layers only (GGG) and four fingerprint

implementations, namely Morgan (boolean), Morgan counts, Morgan chiral (boolean), Morgan chiral counts all with radius 4 and 2048 bits. [38] The fingerprints are embedded by one linear layer with `ReLU` as activation function followed by the aforementioned MLP. More detailed information can be found in Appendix S1.2.

### 3.1.2 Fine-tuning on human feedback

To focus our scoring model on a desired chemical space using human feedback, we apply a fine-tuning approach inspired by Choung et al. [21]. This approach was optimized to require as little data as possible in order to limit the time required from expert chemists labeling the training examples. Datasets to be used for fine-tuning can be of various origins such as a chemical scope suggested by experimentalists or specific chemical spaces encompassing e.g. natural products (see. Section 3.3. Data points from this set of molecules are determined using $k$-means clustering and subsequent pairing of molecules from different clusters. If labels are already available (no human feedback needed) they are used to inform the pairing such that pairs get opposite labels. Subsequently, the uncertainty of the prediction of the pre-trained model on those pairs is determined based on the variance of $\delta_{ij}$ obtained using the Monte Carlo dropout method [39] with a dropout rate of 0.2 on 100 predictions. The top $n$ pairs ($n$ depends on the dataset size) based on the uncertainty are submitted to be evaluated by our expert chemist based on their preference with regard to synthesizability.

These pairs of molecules capturing the desired human intuition or some predefined label (e.g. HS *vs.* ES) are subsequently used for fine-tuning the FSscore model. The fine-tuning process can be achieved with as little as up to 20 epochs. Furthermore, an early stopping approach was designed that incorporates both the improvement on the fine-tuning data while ensuring that the model does not experience degradation of the previously learned. Details can be found in Appendix S1.3.

### 3.2 Data

To pre-train our model on a large collection of reactions we combine the USPTO_full [40] patent dataset with the complementary CJHIF [41]. Reaction data implicitly contains information on the synthetic feasibility through the relation of reactant to product, with the product being synthetically more difficult. The USPTO_full [40] contains reactions extracted from the US patents grants and applications while CJHIF was mined from chemical journals with high impact factor [41]. One data point consists of one reactant and its corresponding product. Reactions with multiple reactants per product were split into separate data points accordingly. The cleaned and filtered dataset was split into training (3,349,455 pairs) and hold-out test set (711,550 pairs) and a random validation set of 1% of the training data was sampled (33,495 pairs). We detail data processing in Appendix S2.1.

### 3.3 Case studies and evaluation

We compare our approach with established scores, namely SA score, SCScore, SYBA and RAscore and report numerous metrics as described in Appendix S3. We investigate the qualitative performance of the pre-trained model on the MOSES [42] and COCONUT [43] datasets. MOSES contains commercially available drug-like molecules and is often used to benchmark generative models for drug discovery and COCONUT is a collection of natural products extracted from various sources. [42, 43]

We showcase the model's ability to efficiently focus our score by fine-tuning on several datasets:

- A subset from the pre-training set with chiral tetrahedral centers. The molecule with assigned chirality is labeled as more complex as opposed to the same molecule stripped of the respective assignment.
- The MC (manually curated) and CP (computationally picked) test sets published with the SYBA score. [23] Labels are provided and correspond to ES and HS.
- The meanComplexity dataset containing averaged complexity scores from chemists. [33] Binary labels are extracted from the continuous score [1,5] based on a set-off of at least two.
- The PROTAC-DB [44], which is an open-source collection of PROTACs. Labels were obtained from human experts.

Additionally, the applicability to a generative modeling task is assessed. REINVENT [7] is used as a molecular generator due to its well-established RL framework and good performance on several

benchmarks. [45] We further make use of the recently proposed augmented memory algorithm for sample efficiency. [46] First, an agent is optimized for docking to the D2 dopamine receptor (DRD2, PDB ID: 6CM4), which is a target for antipsychotic drugs. The collected SMILES are subsequently used to fine-tune the FSscore, which in turn serves as reward for a second round of RL. For comparison, another agent is optimized for the SA score for the same amount of oracle calls. The results are compared by number of reaction steps predicted by AiZynthFinder. [47] All case studies are further detailed in Appendix S2.2.

# 4 Results and discussion

## 4.1 Pre-trained model

The pre-trained model with varying algorithmic and representation implementations was evaluated on the hold-out test set (see Tab. 1). The graph versions outperform fingerprint representations by a small difference and the best-performing model each is used for further investigations. These correspond to the GNN with GATv2 and LineEvo layers (GGLGGL) and the Morgan counts fingerprint. Furthermore, preliminary investigations with models trained on a random subset of 100k data points showed weaknesses of the boolean Morgan fingerprint, resulting in an unfair comparison. Not including the counts of fragments in the fingerprint leads to the inability to capture complexity based on recurrence even if the single fragment is common in the training set. The original publication of the SCScore briefly touches on the impact of varying the Morgan fingerprint, but contrary to us, they found this aspect to not influence the performance, leading them to choose the boolean 1024-bit vector. [22]

Table 1: Performance on the hold-out test set of graph-based models compared to various fingerprint implementations. Accuracy (*Acc*) and *AUC* based on the score differences are reported. The best performing model is highlighted in bold. G = GATv2 layer; L = LineEvo layer

| Model | Representation | *Acc* ↑ | *AUC* ↑ |
|---|---|---|---|
| GNN (GGLGGL) | graph | **0.905** | **0.971** |
| GNN (GGG) | graph | 0.903 | 0.970 |
| MLP | Morgan | 0.87 | 0.954 |
| MLP | Morgan counts | 0.880 | 0.959 |
| MLP | Morgan chiral | 0.867 | 0.952 |
| MLP | Morgan chiral counts | 0.875 | 0.957 |

The analysis of the pre-trained model's predictions on MOSES (known drugs) [42] and COCONUT (natural products) [43] shows that the pre-trained FSscore is unable to distinguish these molecule classes assuming that COCONUT [43] contains more complex structures (see Fig. 1). While both the fingerprint- and graph-based FSscore outperform the SCScore in the area under the receiver operating characteristic curve (*AUC*), the SA score, SYBA and RAscore yield better results. The structures found in COCONUT [43] are likely out-of-distribution for the FSscore highlighting the opportunity for fine-tuning but it also emphasizes the power of rule-based methods such as the SA score.

## 4.2 Fine-tuning

Figure 2 showcases the inability of all baseline scores, including our pre-trained models, to differentiate structures with assigned chirality from those without assignment. Synthesizing a predefined stereoisomer is a much more challenging task than being able to choose or even leave it up to chance. Figure 2 shows that fine-tuning on the chirality test set allows the differentiation of molecules in terms of their chirality assignment resulting in predicting molecules with a given isomer as more difficult to synthesize. The performance could likely be improved by increasing the dataset size as opposed to the 50 pairs used here. While the fingerprint-based models (Morgan *chiral* counts) clearly are able to predict different scores, the meaning of the different representations are not captured by either the pre-trained or fine-tuned model.

Both SYBA test sets show clear room for improvement from the pre-trained baseline, especially on the MC set (see Fig. S4.4). The RAscore, SA score and SYBA clearly outperform the other scores. The RAscore, being an actual classifier, has a structural advantage on this task. The same is true for

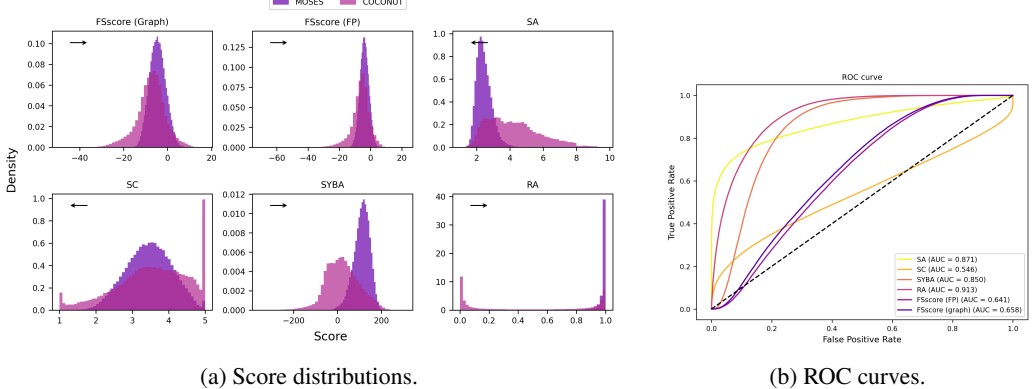

(a) Score distributions.

(b) ROC curves.

Figure 1: Results showcasing the ability to differentiate molecules originating from MOSES [42] from those in COCONUT [43]. The latter are expected to be more complex being natural products. The ROC curves in Figure 1b detail the power to discriminate MOSES [42] from COCONUT [43]. The arrows in the distribution plot (Fig. 1a) indicate the direction of higher synthetic feasibility.

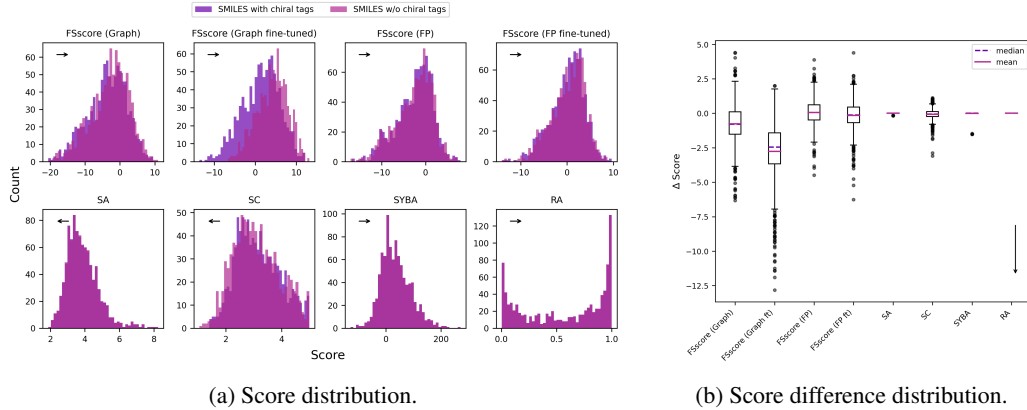

(a) Score distribution.

(b) Score difference distribution.

Figure 2: Distributions showcasing the ability to differentiate molecules with assigned tetrahedral chirality from their unassigned counterpart. The desired prediction would score the assigned molecules as more complex resulting in negative delta values (assigned - unassigned) in Figure 2b.

SYBA, which was purposely trained to perform well on such datasets. The high *AUC* with the SA score is to be expected on the CP set, where all HS molecules contain a ring-bridging atom. The good performance on the MC set is more surprising and showcases how well-formulated rules are valuable on in-distribution datasets. Fine-tuning on molecules from CP resulted in substantial increase in performance yielding an *AUC* of 0.992 when fine-tuning on only 50 pairs. Good performance gain was even achieved with fewer data points as seen in Table S4.2. The MC test set is more challenging and the limited dataset size (40 pairs) makes evaluation challenging. However, Table S4.2 clearly shows that the FSscore can also improve such a heterogeneous dataset. When using overlapping pairs (molecules can appear in multiple pairs) the performance is improved substantially improved over the unique setting, however with the trade-off of reduced generalizability measured as performance on the pre-training test set (see Table S4.3). Furthermore, our goal is to keep the fine-tuning size to a minimum in order to facilitate human labeling.

The ability to score molecules, whose complexity was assigned by chemists was assessed on the meanComplexity dataset by Sheridan et al. [33]. The correlations between all scores and the meanComplexity are shown in Figure S4.6. The correlation to meanComplexity was improved through fine-tuning (see Table S4.4) but still lacks behind scores such as SA score or RAscore. This could likely be rescued by increasing the fine-tuning dataset size from 50 pairs, which we aimed at keeping small. Only by using 500 overlapping pairs (molecules can appear in multiple pairs) can the fine-tuned FSscore outperform the SA score in terms of PCC (0.84 *vs.* 0.8 – see Figure S4.7).

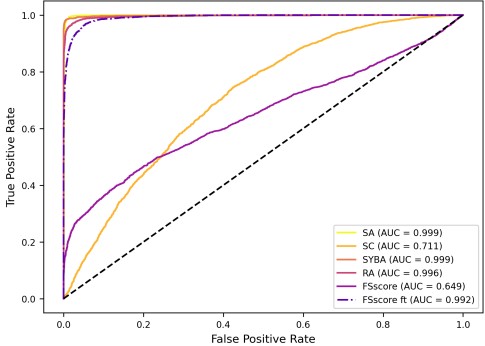
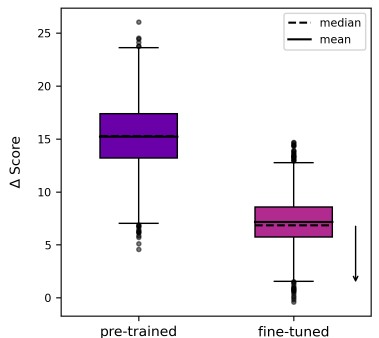

Figure 3: ROC curves showcasing the classification power of the various models to separate HS from ES in the CP test set.

Figure 4: Score difference between full PROTAC and most complex respective fragment (either of the two ligands or linker).

PROTACs are large molecules (700-1100 Da) compared to traditional drugs but their fragmented composition allows synthesis of each ligand and the linker separately before connecting the three parts. Thus, their size often "fools" known scores, which rank them as synthetically hard making their relative scores unreliable. We obtain feedback from a chemist with expertise on PROTACs, who also remarked that most molecules they were prompted with are relatively easy to make. Figure 5 shows a distinct shift towards higher predicted synthetic feasibility after fine-tuning the FSscore and Figure 4 highlights the narrowing gap between each PROTAC and the respective most complex fragment (ligands or linker) after fine-tuning. The latter observation is desirable under the assumption that, given that connecting the three fragments is not challenging, the complexity of the full PROTAC is not much higher than its most complex components. However, the increase in performance is small (50 pairs: *Acc* increases from 0.53 to 0.57 and *AUC* from 0.43 to 0.52) on all tested fine-tuning dataset sizes and evaluation is challenging with only 100 labels (see Tab. S4.2). The performance gain on all PROTACs and the learning curves (see Fig. S6.29) clearly show the ability to learn relevant features emphasizing the need for more labeled data.

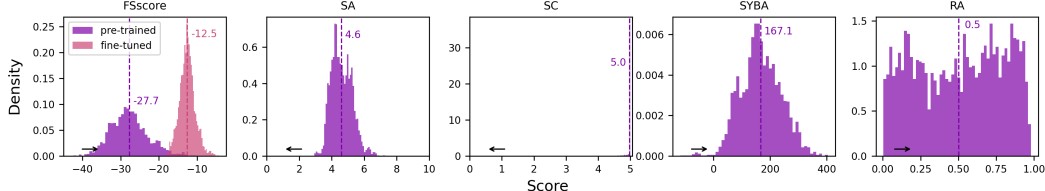

Figure 5: Score distributions obtained for the PROTAC-DB sample. The first plot shows the scores before (grey) and after (purple) fine-tuning on 80 pairs of human-labeled PROTACs. The arrows each indicate direction of higher synthetic feasibility and the dashed line marks the mean.

Our last case study couples human feedback to a generative model attempting targeted improvement of synthetic feasibility of the generated molecules. The optimization of the first agent for docking to DRD2 leads to the generation of lipophilic structures and polyaromatic ring systems. To improve the synthesizability a second agent is optimized for the fine-tuned FSscore. Figure 6 shows the superiority of the FSscore over the SA score in capturing the synthetic feasibility and generating molecules that are predicted to require less reaction steps. However, this comes at the expense in terms of docking score compared to the SA-optimized agent, which in turn generated many outliers with high docking scores. Figure S5.21 shows examples of generated structures of all three agents.

## 5   Limitations and future work

The pre-trained model often achieves worse results than the SA score indicating that the incorporation of carefully selected rules could be beneficial for a baseline. While focusing the score towards the chemical space of interest is attractive for many applications, it also poses its challenges to keep

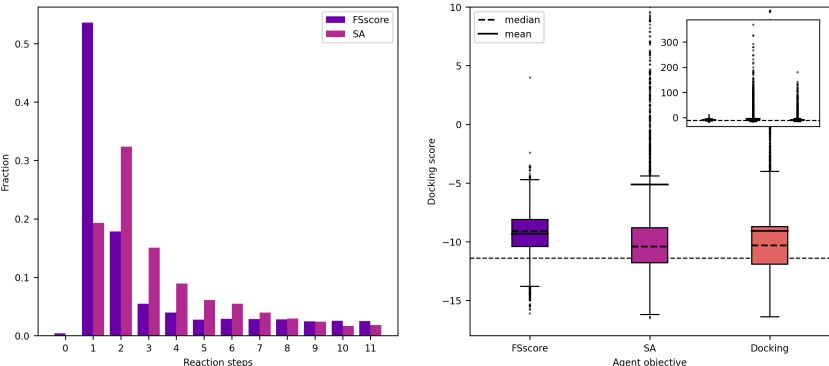

Figure 6: *Left:* Fraction of reaction steps predicted by AiZynthFinder for each generated molecule. Zero reaction steps indicate that the structure was found in the buyable stocks. *Right:* Box plots displaying the distribution of docking scores for the three trained agents. The dashed line indicates the docking score of the Risperidone that is the true ligand present in the used crystal structure.

baseline generalizability. Furthermore, the necessity for individual fine-tuning comes at the expense of convenience as the FSscore is not meant to be used out of the box. The dataset size required for fine-tuning seems to be related to the complexity and homogeneity of the data. However, recruiting human labelers with sufficient expertise can be a challenge on its own and limited labels makes evaluation challenging. Further investigations should aim at performing a more extensive hyperparameter search as well as optimizing the architecture itself. Furthermore, a study of larger scale with more expert chemists would be of benefit to showcase generalizability and robustness. Lastly, extending the use for generative models in a variety of downstream tasks would be beneficial. Such experiments should include different architectures, a variety of specific objectives as well as ways of incorporating the score potentially through iterative fine-tuning in a active learning framework. It can be argued that directly incorporating synthesizability by design leads to more robust results in terms of synthetic feasibility but it also restricts the accessible chemical space more. [20]

# 6    Conclusion

This work introduces the novel FSscore, for evaluating synthetic feasibility, leveraging pairwise preferences and fine-tuning with human feedback to focus on specific chemical spaces. The score could be applied as filter in VS studies or as reward in RL frameworks and being fully differentiable allows the seamless integration into generative models. The use of pre-training on an extensive reaction dataset establishes a robust baseline, implicitly capturing synthetic complexity. Importantly, our experiments demonstrate the practicality of FSscore, showcasing its efficacy even with very small amounts of labeled data in certain cases. Fine-tuning improved performance on several chemical scopes over the pre-trained baseline, demonstrating the approach's ability to adapt to new domains.

# 7    Code availability and app

All code used for training, fine-tuning and scoring the FSscore is available at `https://github.com/schwallergroup/fsscore`. This repository also included an application that can be run locally and allows a more intuitive and accessible way to label data, fine-tune and deploy a model.

## Acknowledgments and Disclosure of Funding

RMN thanks VantAI (USA) for support. PS acknowledges support from the NCCR Catalysis (grant number 180544), a National Centre of Competence in Research funded by the Swiss National Science Foundation. We also thank Chalada Suebsuwong and Zlatko Jončev for their expert chemistry feedback and Théo Neukomm for his help with the application.

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

# Supplementary Information

## S1 Training details

### S1.1 Initial featurization of graphs

Table S1.1: Features for initializing the nodes and edges of the graph. One-hot encoding includes one additional bit for types not found in the choices (incl. in size). These properties were determined using RDKit. rdk [48]

| Localization | Feature | Description | Size |
|---|---|---|---|
| atom | atom type | one-hot encoded atom type [1] | 14 |
| | charge | one-hot encoded formal charge [-4,4] | 10 |
| | implicit hydrogens | one-hot encoded number of hydrogens [0,4] | 6 |
| | degree | one-hote encoded degree [0,4] | 6 |
| | ring information | 1 if in ring else 0 | 1 |
| | aromaticity | 1 if aromatic else 0 | 1 |
| | hybridization | one-hot encoded hybridization state [2] | 9 |
| | chiral tag | one-hot encoded chiral tag [S, R, unassigned] | 4 |
| bond | bond type | one-hot encoded bond type [3] | 5 |
| | conjugated | 1 if conjugated else 0 | 1 |
| | ring information | 1 if in ring else 0 | 1 |

### S1.2 Pre-training

The GNN operates on a hidden size of 128 and the input dimensions depend on the initial featurization of the edges and nodes (see Tab. S1.1. GATv2 layers use 8 heads and LeakyReLU as activation function after updating the hidden feature vector while LineEvo layers use ELU. The updated averaged node representation is transformed using PReLU in GATv2 layers and with ELU in LineEvo layers. After concatenating the global max pooled and global weight-add pooled node representations to a molecular representation for every layer (G or L) this new representation is passed through an MLP with 3 hidden layers of size 256 and ReLU as activation function. The fingerprint-based models have an encoder of an MLP of one hidden layer of size 256 (input dimension 2048) followed by the same MLP of 3 layers as above.

The pre-trained model was trained with the Adam optimizer [49] and an initial learning rate of 3e-4. To center the obtained score around zero a regularization factor of 1e-4 was applied to the predicted score like to obtain the regularization loss $\mathcal{L}_{reg}(\hat{f}; \lambda) := \lambda ||\hat{f}||^2$ as suggested by Choung et al. [21]. $\mathcal{L}_{reg}$ is added to the cross-entropy loss term as described in Section 3.1.1 to obtain the final loss $\mathcal{L} = \mathcal{L}_{CE} + \mathcal{L}_{reg}$. The training set was randomly split in 25 equally sized subsets and the model was trained with each subset for 10 epochs totalling 250 epochs with a batch size of 128. This sequential learning approach was found to work well to increase speed of training.

### S1.3 Fine-tuning

Training our model requires pairs of molecules with a binary preference as label. In order to do so, we first cluster a dataset using the k-means algorithm based on the Tanimoto distances of Morgan counts fingerprint. The number of clusters k was determined for every dataset individually where the mean Silhouette coefficient over a range of k was the smallest. [50]. The range queried was set from 5 to 29 for datasets larger than 500 pairs and to 3 to 9 for smaller datasets. Secondly, molecules are paired in such a way that they come from different clusters, appear only once in the paired dataset and have opposite labels if available. We also investigated the influence of having overlapping pairs

---

[1] Considered atom types: [H, B, C, N, O, F, Si, P, S, Cl, Br, I, Se]
[2] Considered hybridization states: [UNSPECIFIED, s, sp, sp2, sp3, sp3d, sp3d2, other]
[3] Considered bond types:[single, double, triple, aromatic]

(molecules can appear multiple times) for the SYBA (CP and MC) and the meanComplexity datasets. This approach results in more fine-tuning data but does not satisfy our desire to keep the dataset size small since human labeling could not make use of this advantage. Subsequently, the uncertainty of the prediction of the pre-trained model on those pairs is determined based on the variance of $\delta_{ij}$ obtained using the Monte Carlo dropout method [39] with a dropout rate of 0.2 on 100 predictions. The dataset is sorted with descending variance so that the pairs with high uncertainty are labeled first or used first for fine-tuning when selecting a subset. If no label is available the pairs get subsequently labeled by a chemist.

We chose a transfer learning type of fine-tuning where all the weights of the pre-trained model can get adapted. All the hyperparameter except learning rate and batch size are kept the same. In order to avoid the forgetting the previously learned we track the performance metrics on a random subset of 5,000 pairs from the hold-out test set of the pre-training data. Of the four tested initial learning rates (1e-5, 1e-4, 3e-4, 1e-3) we chose 1e-4 for the graph-based version and 3e-4 for the fingerprint-based models. These learning rates are a good trade-off between degradation of the previously learned and learning on the new dataset as can be seen in the learning curves in Appendix S4.1. We observed quick conversion with fewer than 20 epochs and in order to balance the aforementioned trade-off a custom early stopping method was applied. Training is stopped once either the validation loss (or training loss if the full dataset is used for production) has increased for 3 epochs (patience of 3) or the accuracy on the 5k subset of the hold-out test set decreased for 3 epochs with a delta threshold of 0.02. We trained for a maximum of 20 epochs. For evaluating the efficiency, we further varied the number of data points used for training of which we took 5 random pairs for validation. The results in Table S4.2 show improvement in performance the bigger the dataset size. The batch size was set to 4.

## S2 Data

### S2.1 Pre-training data

The USPTO_full dataset was downloaded according to `https://github.com/coleygroup/Graph2SMILES/blob/main/scripts/download_raw_data.py` [51] using all USPTO_full subsets and "src" as product and "tgt" as reactants. The dataset was split so that every data point is a pair of one reactant and its respective product and the SMILES strings were canonicalized. The datasets do not contain reactions with multiple products. Data points with empty strings, identical reactants and products (isomeric SMILES), either reactant or product containing less than four heavy atoms or containing element types not in {H, B, C, N, O, F, Si, P, S, Cl, Se, Br, I} were removed. The datasets were further deduplicated retaining one instance of replicates. Furthermore, data points were removed ensuring that there are no cycles in the reaction network. This was achieved by removing back edges with the Depth-First Search (DFS) algorithm as implemented by Sun et al. [52]. The filtered dataset consisted of 5,340,704 data points. The train test split was performed so that no molecules (note not just reactant:product pairs) are overlapping resulting in a significant data loss to prevent data leakage. The training set consists of 3,349,455 pairs and the hold out test set of 711,550 pairs (17.5%).

To qualitatively evaluate the pre-trained model the performance on the MOSES [42] and CO-CONUT [43] dataset were assessed. We downloaded the MOSES test set from Polykovskiy et al. [42] as is totalling 176,074 SMILES. COCONUT was extracted from Sorokina et al. [43], all SMILES returning `None` with `RDKit` [48] were removed and a random subset of 176,000 molecules was selected to match the MOSES fraction.

### S2.2 Fine-tuning data

The data for the fine-tuning case studies come from various sources. The specific processing steps for every dataset is outlined below.

The obtain a test set for the chirality case study we filtered the deduplicated molecules from the pre-training training set keeping only those with `@` or `@@` tokens in the SMILES string, which define the chirality of a tetrahedral stereocenter. Of those we sample 1,000 SMILES randomly and obtain their partner by getting the non-isomeric SMILES using `RDKit` rdk [48].

The CP and MC test sets were extracted from Voršilák et al. [23]. No further processing was required.

The meanComplexity dataset was downloaded from Sheridan et al. [33] and cleaned by removing SMILES whose conversion to an `RDKit` molecule object failed. This processing step removed 44 SMILES yielding a fine-tuning test set of 1,731 molecules.

For the PROTAC case study we extracted all PROTAC SMILES from the PROTAC-DB Weng et al. [44] and deduplicated the dataset keeping the first instance resulting in 3,270 SMILES. After pairing and ranking by confidence as described in Appendix S1.3, 100 pairs were labeled by a medicinal chemist with expertise on PROTACs. In order to compare the scores of the individual components to the full PROTAC as done in Figure 4 the individual fragments (the two ligands and the linker) has to be extracted. For this, we extracted the collections containing information on anchor and warhead (the two ligands) from PROTAC-DB and tried to match an anchor and warhead to every PROTAC by matching protein target ID and find the biggest substructure match. The linker was obtained by removing the ligands' atoms. For the 3270 PROTACs in the database we could find matching ligands for 1920 of them. This approach of extraction is necessary because PROTAC-DB does not cross-reference the PROTACs to the fragments and our extraction cannot guarantee concordance with the reported warhead and anchor in the respective publications but still is a good approximation.

To assess the applicability of the FSscore to a generative task we took advantage of the RL-framework of REINVENT [7] and used the recently proposed augmented memory [46] optimization protocol. First, an agent was trained with the composite (equal weights) of the docking score to the D2 Dopamine receptor (DRD2, PDB ID: 6CM4) and the molecular weight (MW) as reward. The docking score was determined using AutoDock Vina [53] on one conformer each. Conformer generation is performed with `RDKit` [48] and the Universal force field (UFF) [54] for energy minimisation for a maximum of 600 iterations. The score $x$ was scaled to obtain reward $x'$ in a range of [0,1] by applying a sigmoid transformation as followed:

$$x' = \frac{1}{1 + 10^{10k \cdot \frac{x - \frac{a+b}{2}}{a-b}}} \tag{S2.1}$$

with $a$ corresponding to a docking score $x$ of -1, $b$ -13 and the steepness $k$ was set to 0.25 resulting in a reverse sigmoid (a small docking score results in a reward $x'$ towards 1 while a high score returns a reward $x'$ close to 0). The score for MW was formulated so that a high reward is returned when having a MW between 0 and 500 Da using a double sigmoid like so:

$$\tag{S2.2}$$

$$A = 10^{c_{SE} \frac{x}{c_{div}}} \tag{S2.3}$$

$$B = 10^{c_{SE} \frac{x}{c_{div}}} + 10^{c_{SE} \frac{b}{c_{div}}} \tag{S2.4}$$

$$C = \frac{10^{c_{SI} \frac{x}{c_{div}}}}{10^{c_{SI} \frac{x}{c_{div}}} + 10^{c_{SI} \frac{a}{c_{div}}}} \tag{S2.5}$$

$$x' = \frac{A}{B} - C \tag{S2.6}$$

with coefficients $c_{SE} = c_{SI} = 500$ and $c_{div} = 250$, the lower bound $a = 0$ and upper bound $b = 500$. RL was carried out for 150 epochs, a batch size of 64, learning rate of 1e-4, a $\sigma$ of 128 and augmented memory as optimization algorithm with 2 augmentation rounds and selective memory purge with a minimum similarity of 0.4 and a bin size of 10 based on the identical Murcko scaffold. For details on augmented memory in REINVENT we refer to Guo and Schwaller [46]. All valid SMILES (9,246) generated during this optimization process were subsequently used to fine-tune the FSscore as described above. The agent already optimized for docking was next used to optimize for synthetic feasibility either with the fine-tuned FSscore or the SA score as comparison. Both approaches were trained in identical fashion with the following parameters: 50 epochs, batch size of 128, $\sigma$ of 128, a learning rate of 1e-4 and using the augmented memory algorithm with 2 rounds of augmentation. The diversity filter (incl. selective memory purge) was based on identical Murcko scaffolds based on a minimal similarity of 0.4, a minimal score of 0.4 and a bucket size of 25. The FSscore was transformed to range from 0 to 1 with a sigmoid according to Equation S2.1 with $a = -13.08$, $b = 10.07$ and a steepness $k = 0.25$. The agent optimized for the SA score was transformed similarly reverting the sigmoid by setting $a = 10$, $b = 1$ and $k = 0.25$. Thus, a high FSscore result in a high reward, while a low SA score results in high reward and vice versa. As mentioned above, only molecules with a minimal score of 0.4 were collected and subsequently used for evaluation. This yielded 4,779 SMILES from the FSscore-trained agent and 4,915 SMILES from the SA score-trained agent.

## S3 Metrics

The Pearson correlation coefficient (*PCC*) between $x$ and $y$ is computed using the function `scipy.stats.pearsonr` [55] and is calculated as followed:

$$PCC = \frac{\sum(x - m_x)(y - m_y)}{\sqrt{\sum(x - m_x)^2 \sum(y - m_y)^2}} \tag{S3.7}$$

with $m_x$ the mean of vector $x$ and $m_y$ the mean of vector y.

The following metrics for datasets with binary labels (e.g. HS *vs.* ES) are computed using the `sklearn.metrics` package [56]. Accuracy (*Acc*), sensitivity (*SN*) and specificity are calculated based on the relative sizes of true positives (*TP*), true negatives (*TN*), false positives (*FP*) and false negatives (*FN*):

$$Acc = \frac{TP + TN}{TP + TN + FP + FN} \tag{S3.8}$$

$$SN = \frac{TP}{TP + FN} \tag{S3.9}$$

$$SP = \frac{TN}{TN + FP} \tag{S3.10}$$

The area under the receiver operating characteristic (ROC) curve (*AUC*) describes the ability to discriminate data points based on binary labels at various thresholds. It is computed by plotting *SN* against $1 - SP$ at various cut-off values defining the two classes. A random classifier has an *AUC* of 0.5.

## S4 Additional results

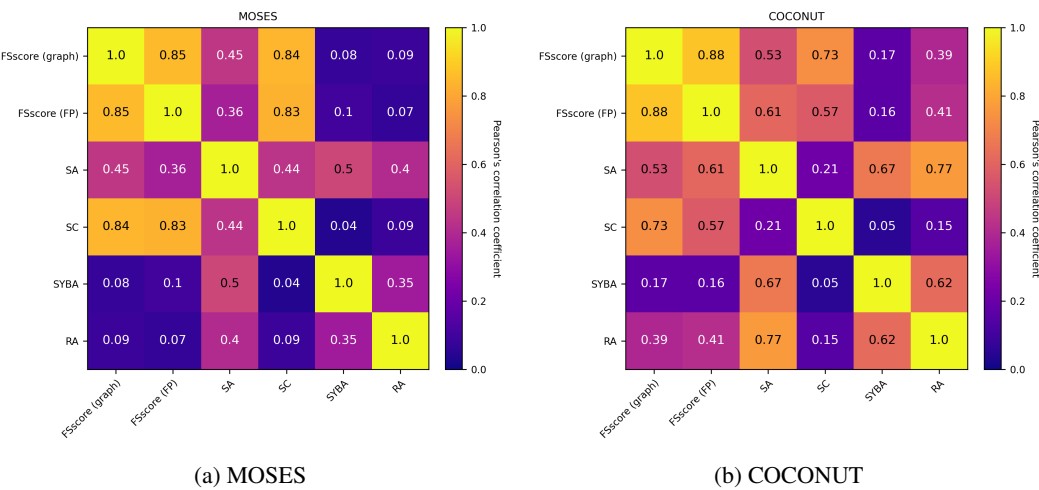

(a) MOSES

(b) COCONUT

Figure S4.1: Heat maps displaying the correlations (*PCC*) between all scores obtained on molecules from MOSES and COCONUT.

## S4.1 Fine-tuning results

Table S4.2: Performance metrics showcasing the improvement on specific datasets after fine-tuning at different fine-tuning dataset sizes ($p_{ft}$ pairs). No FT refers to the performance of the pre-trained model on those datasets. $Acc_{pt}$ and $AUC_{pt}$ are determined on the pre-training test set and are based on the score difference (as during training) not the score itself. The values for the fine-tuned versions always show metrics on the dataset excluding the training molecules and the full dataset in brackets.

| set | mode | $p_{ft}$ | Acc | AUC | $Acc_{pt}$ | $AUC_{pt}$ |
|---|---|---|---|---|---|---|
| chiral | | no FT | 0.5435 | 0.5391 | 0.905 | 0.971 |
| | graph | 20 | 0.5762 (0.575) | 0.5972 (0.5947) | 0.9044 | 0.9706 |
| | | 30 | 0.5918 (0.592) | 0.6185 (0.6173) | 0.9035 | 0.97 |
| | | 40 | 0.6156 (0.6145) | 0.6539 (0.6521) | 0.9022 | 0.9692 |
| | | 50 | 0.6268 (0.6235) | 0.6749 (0.6728) | 0.8983 | 0.9671 |
| | | no FT | 0.509 | 0.4989 | 0.875 | 0.957 |
| | fp | 20 | 0.5163 (0.5165) | 0.5082 (0.5102) | 0.879 | 0.959 |
| | | 30 | 0.5180 (0.5195) | 0.5122 (0.5153) | 0.8786 | 0.9583 |
| | | 40 | 0.5255 (0.5255) | 0.5180 (0.5212) | 0.8738 | 0.9544 |
| | | 50 | 0.5237 (0.5235) | 0.5147 (0.5174) | 0.8783 | 0.9582 |
| CP [23] | | no FT | 0.6319 | 0.6446 | 0.905 | 0.971 |
| | graph | 20 | 0.9275 (0.9273) | 0.9804 (0.9802) | 0.903 | 0.9700 |
| | | 30 | 0.943 (0.9429) | 0.9868 (0.9865) | 0.9021 | 0.9695 |
| | | 40 | 0.9592 (0.9587) | 0.9933 (0.9930) | 0.9012 | 0.9689 |
| | | 50 | 0.9567 (0.9560) | 0.9921 (0.9918) | 0.9008 | 0.9687 |
| | | no FT | 0.7008 | 0.7566 | 0.880 | 0.959 |
| | fp | 20 | 0.8683 (0.8685) | 0.9396 (0.9397) | 0.8771 | 0.9568 |
| | | 30 | 0.8833 (0.8838) | 0.9517 (0.9521) | 0.8741 | 0.9543 |
| | | 40 | 0.9043 (0.9045) | 0.9645 (0.9649) | 0.8732 | 0.9536 |
| | | 50 | 0.9167 (0.9178) | 0.9725 (0.9731) | 0.8708 | 0.952 |
| MC [23] | | no FT | 0.5375 | 0.4894 | 0.905 | 0.971 |
| | graph | 20 | 0.65 (0.7875) | 0.645 (0.7744) | 0.9034 | 0.9695 |
| | | 30 | 0.6 (0.7625) | 0.55 (0.8150) | 0.9034 | 0.9696 |
| | | 40 | (0.85) | (0.906) | 0.8998 | 0.9679 |
| | | no FT | 0.5875 | 0.515 | 0.880 | 0.959 |
| | fp | 20 | 0.6 (0.7125) | 0.5425 (0.7832) | 0.8820 | 0.9596 |
| | | 30 | 0.7 (0.8875) | 0.61 (0.8994) | 0.8734 | 0.9529 |
| | | 40 | (0.825) | (0.8513) | 0.8786 | 0.9579 |
| PROTAC-DB [44] | | no FT | 0.53 | 0.4341 | 0.905 | 0.971 |
| | graph | 20 | 0.5063 (0.515) | 0.4341 (0.4602) | 0.905 | 0.9715 |
| | | 30 | 0.5214 (0.555) | 0.4587 (0.5474) | 0.9023 | 0.9697 |
| | | 40 | 0.5083 (0.51) | 0.4583 (0.4781) | 0.9048 | 0.9712 |
| | | 50 | 0.57 (0.6) | 0.522 (0.6142) | 0.8993 | 0.968 |
| | | 60 | 0.55 (0.615) | 0.5097 (0.6389) | 0.8984 | 0.9672 |
| | | 70 | 0.5167 (0.675) | 0.4433 (0.6857) | 0.8954 | 0.9659 |
| | | 80 | 0.55 (0.685) | 0.4725 (0.7281) | 0.8975 | 0.9672 |
| generated | | no FT | 0.5594 | 0.5255 | 0.905 | 0.971 |
| | graph | 20 | 0.5741 (0.5842) | 0.5437 (0.5629) | 0.8971 | 0.9675 |
| | | 30 | 0.5775 (0.6139) | 0.5444 (0.6117) | 0.8969 | 0.9673 |
| | | 40 | 0.5656 (0.6535) | 0.5612 (0.6686) | 0.8942 | 0.9659 |
| | | 50 | 0.5882 (0.604) | 0.5692 (0.589) | 0.9012 | 0.9694 |
| | | 60 | 0.5823 (0.6485) | 0.5756 (0.6566) | 0.8887 | 0.9632 |
| | | 70 | 0.6034 (0.6287) | 0.6177 (0.6527) | 0.8907 | 0.9643 |
| | | 80 | 0.6316 (0.7376) | 0.6357 (0.7752) | 0.8847 | 0.9616 |

Table S4.3: Performance metrics showcasing the improvement on specific datasets after fine-tuning at different fine-tuning dataset sizes with overlapping pairs (molecules can appear in multiple pairs). No FT refers to the performance of the pre-trained model on those datasets. $Acc_{pt}$ and $AUC_{pt}$ are determined on the pre-training test set and are based on the score difference (as during training) not the score itself. The values for the fine-tuned versions always show metrics on the dataset excluding the training molecules (with evaluation size $n_{eval}$) and the full dataset in brackets.

| set | mode | $p_{ft}$ | $n_{eval}$ | Acc | AUC | $Acc_{pt}$ | $AUC_{pt}$ |
|-----|------|----------|------------|-----|-----|------------|------------|
| CP [23] | | no FT | 7162 | 0.6319 | 0.6446 | 0.905 | 0.971 |
| | graph | 20 | 7122 | 0.8869 (0.8869) | 0.9543 (0.9544) | 0.9037 | 0.9639 |
| | | 50 | 7062 | 0.9598 (0.9598) | 0.9933 (0.9933) | 0.9008 | 0.9622 |
| | | 100 | 6962 | 0.9779 (0.9779) | 0.9978 (0.9978) | 0.8984 | 0.9609 |
| | | 500 | 6162 | 0.9982 (0.9982) | 1.0 (1.0) | 0.8773 | 0.9469 |
| | | 1000 | 5163 | 0.9983 (0.9983) | 1.0 (1.0) | 0.8662 | 0.941 |
| | | 2000 | 3371 | 0.9990 (0.9990) | 1.0 (1.0) | 0.8497 | 0.9318 |
| | fp | no FT | 7162 | 0.7008 | 0.7566 | 0.880 | 0.959 |
| | | 20 | 7122 | 0.8489 (0.8489) | 0.9201 (0.9207) | 0.8761 | 0.9471 |
| | | 50 | 7062 | 0.8833 (0.8830) | 0.9517 (0.9487) | 0.8721 | 0.944 |
| | | 100 | 6962 | 0.9102 (0.9102) | 0.9679 (0.9689) | 0.8688 | 0.9403 |
| | | 500 | 6162 | 0.9789 (0.9789) | 0.9970 (0.9974) | 0.8564 | 0.9285 |
| | | 1000 | 5164 | 0.9901 (0.9901) | 0.9989 (0.9993) | 0.8415 | 0.9168 |
| | | 2000 | 3372 | 0.9965 (0.9965) | 0.9997 (0.9999) | 0.8179 | 0.8981 |
| MC [23] | | no FT | 80 | 0.5375 | 0.4894 | 0.905 | 0.971 |
| | graph | 20 | 59 | 0.6625 (0.6625) | 0.5829 (0.6656) | 0.8985 | 0.9607 |
| | | 50 | 41 | 0.775 (0.775) | 0.7273 (0.8175) | 0.8971 | 0.9593 |
| | | 100 | 32 | 0.8125 (0.8125) | 0.8155 (0.8244) | 0.8995 | 0.9609 |
| | | 500 | 20 | 0.9 (0.9) | 0.6813 (0.9375) | 0.8698 | 0.9414 |
| | fp | no FT | 80 | 0.5875 | 0.515 | 0.880 | 0.959 |
| | | 20 | 60 | 0.6375 (0.6375) | 0.4733 (0.6313) | 0.8797 | 0.9502 |
| | | 50 | 44 | 0.85 (0.85) | 0.7789 (0.9106) | 0.8669 | 0.9415 |
| | | 100 | 29 | 0.925 (0.9250) | 0.8389 (0.9484) | 0.8545 | 0.9312 |
| | | 500 | 20 | 0.95 (0.95) | 0.8542 (0.9612) | 0.8324 | 0.9118 |

### S4.1.1 CP test set

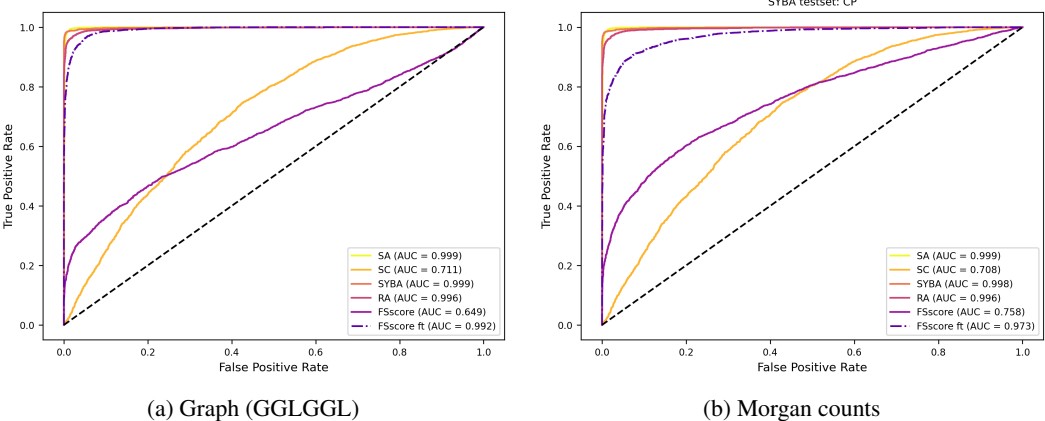

(a) Graph (GGLGGL)  (b) Morgan counts

Figure S4.2: ROC curves showcasing the ability to distinguish HS from ES in the CP test set using the graph-based FSscore or the fp-based FSscore. The fine-tuning was done with 50 pairs and these 100 molecules were excluded from the plots.

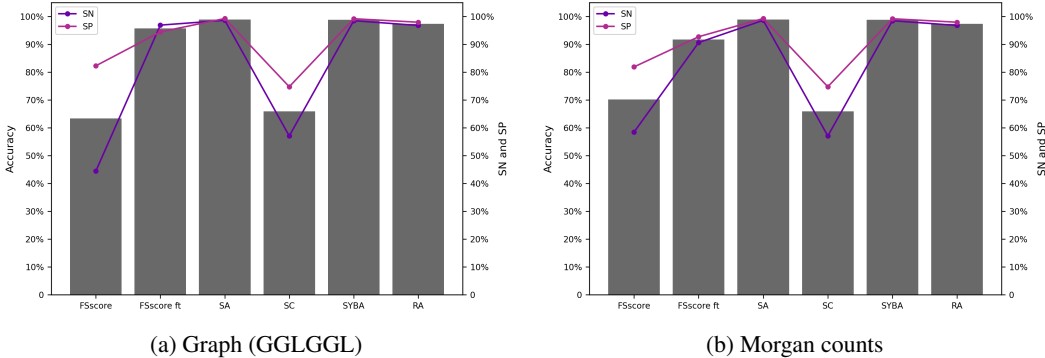

(a) Graph (GGLGGL)  (b) Morgan counts

Figure S4.3: Summary of *Acc*, *SN* and *SP* for classifying the molecules from the CP test set as either HS or ES. The fine-tuning was done with 50 pairs and these 100 molecules were excluded from the plots.

### S4.1.2 MC test set

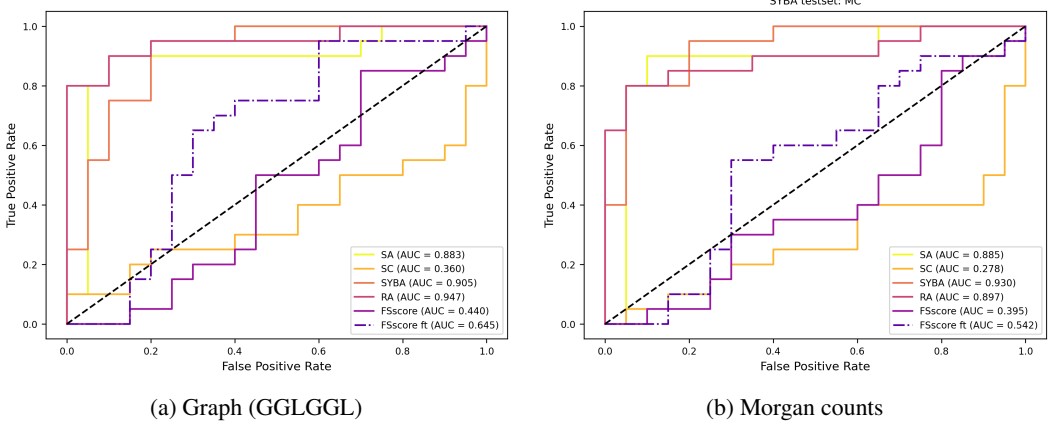

(a) Graph (GGLGGL)                           (b) Morgan counts

Figure S4.4: ROC curves showcasing the ability to distinguish HS from ES in the MC test set using the graph-based FSscore or the fp-based FSscore. The fine-tuning was done with 20 pairs and these 40 molecules were excluded from the plots.

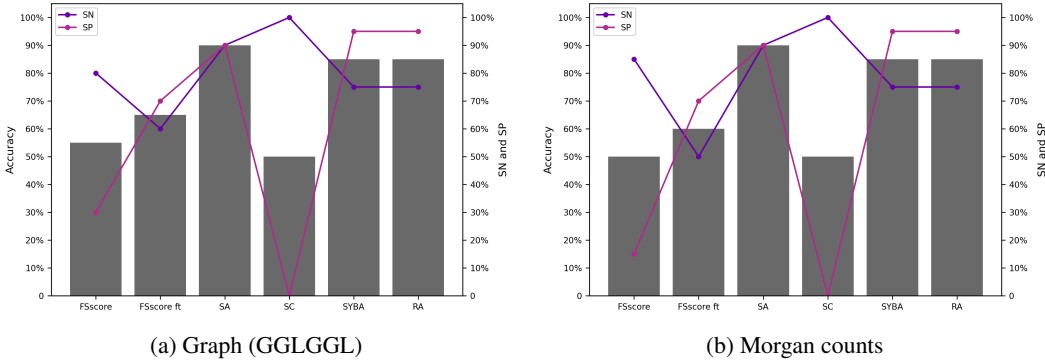

(a) Graph (GGLGGL)                           (b) Morgan counts

Figure S4.5: Summary of *Acc*, *SN* and *SP* for classifying the molecules from the MC test set as either HS or ES. The fine-tuning was done with 20 pairs and these 40 molecules were excluded from the plots.

## S4.2 meanComplexity dataset

Table S4.4: Performance metrics showcasing the improvement on the meanComplexity dataset (Sheridan et al. [33]) after fine-tuning at different fine-tuning dataset sizes with (unique = yes) or without overlapping pairs (molecules can appear in multiple pairs). No FT refers to the performance of the pre-trained model on those datasets. $Acc_{pt}$ and $AUC_{pt}$ are determined on the pre-training test set and are based on the score difference (as during training) not the score itself. The values for the fine-tuned versions always show metrics on the dataset excluding the training molecules (with evaluation size $n_{eval}$) and the full dataset in brackets.

| dataset | unique | mode | $p_{ft}$ | $n_{eval}$ | PCC | $Acc_{pt}$ | $AUC_{pt}$ |
|---|---|---|---|---|---|---|---|
| meanComplexity [33] | yes | | no FT | 1681 | 0.51 | 0.905 | 0.971 |
| | | graph | 20 | 1641 | 0.61 | 0.9034 | 0.9701 |
| | | | 30 | 1621 | 0.63 | 0.9035 | 0.9697 |
| | | | 40 | 1601 | 0.66 | 0.9033 | 0.9697 |
| | | | 50 | 1581 | 0.66 | 0.9028 | 0.9689 |
| | | fp | no FT | 1681 | 0.49 | 0.880 | 0.959 |
| | | | 20 | 1641 | 0.58 | 0.8768 | 0.9569 |
| | | | 30 | 1621 | 0.68 | 0.8668 | 0.9492 |
| | | | 40 | 1601 | 0.68 | 0.8483 | 0.9353 |
| | | | 50 | 1581 | 0.67 | 0.8674 | 0.9502 |
| | no | graph | no FT | 1681 | 0.51 | 0.905 | 0.971 |
| | | | 20 | 1649 | 0.55 | 0.8995 | 0.9615 |
| | | | 50 | 1582 | 0.51 | 0.9017 | 0.963 |
| | | | 100 | 1552 | 0.73 | 0.8921 | 0.9564 |
| | | | 500 | 1291 | 0.84 | 0.8765 | 0.9447 |
| | | fp | no FT | 1681 | 0.49 | 0.880 | 0.959 |
| | | | 20 | 1648 | 0.51 | 0.8809 | 0.9513 |
| | | | 50 | 1584 | 0.59 | 0.8763 | 0.9475 |
| | | | 100 | 1538 | 0.7 | 0.8632 | 0.9377 |
| | | | 500 | 1292 | 0.8 | 0.8412 | 0.9212 |

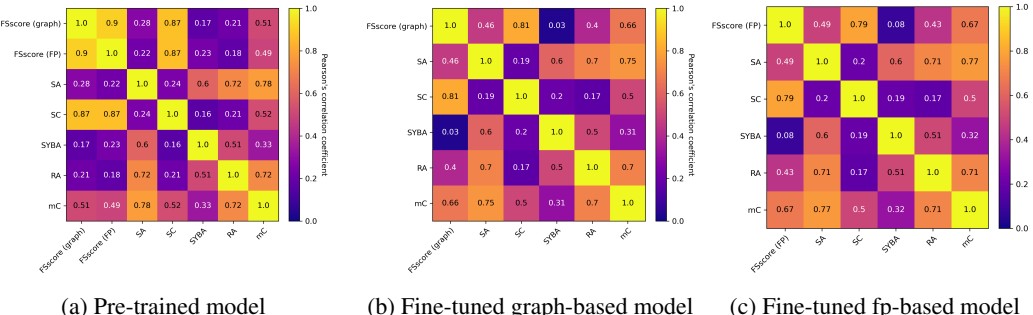

(a) Pre-trained model     (b) Fine-tuned graph-based model     (c) Fine-tuned fp-based model

Figure S4.6: Heat maps displaying the correlations (PCC) between all scores including the meanComplexity (mC) obtained on molecules from the meanComplexity dataset. The plots based on fine-tuned models excluded the fine-tuning data (50 pairs).

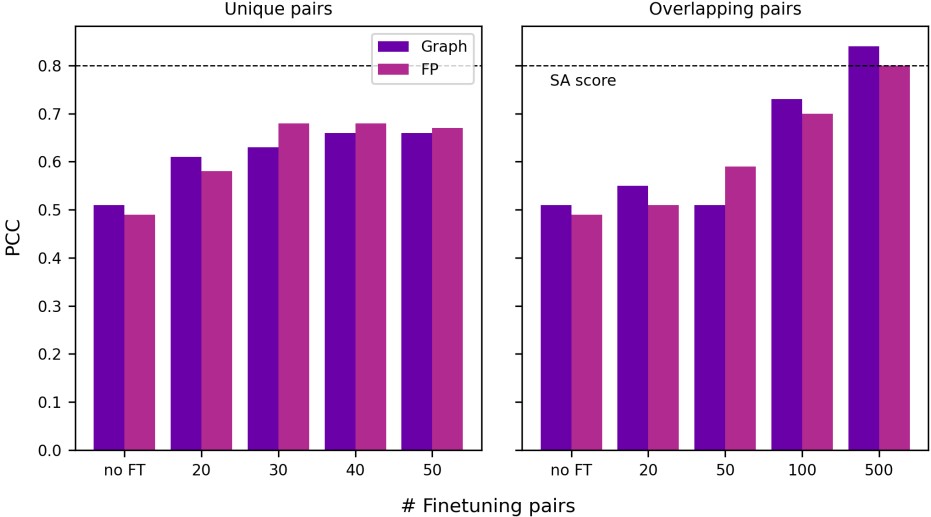

Figure S4.7: Correlations (PCC) between the meanComplexity and the FSscore fine-tuned with different dataset sizes and with unique (*left*) or overlapping (*right*) pairs for both graph and fingerprint (FP) representations. The dashed line indicates the PCC of the SA score to the meanCompelxity being the best performing score we have tested.

# S5 Example structures

## S5.1 Hold-out test set

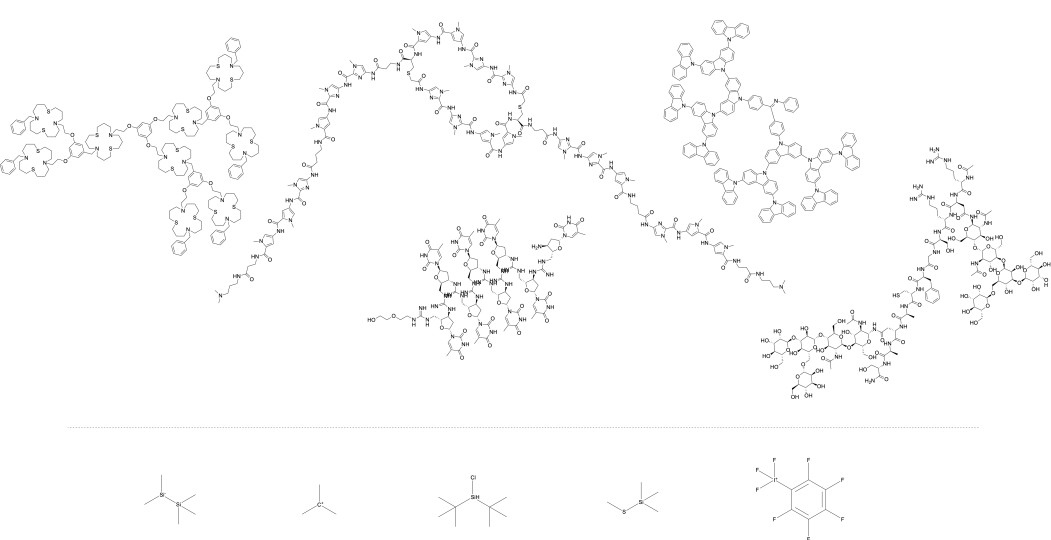

Figure S5.8: Structures from the hold-out test set scored by the pre-trained graph (**GGLGGL**) model. *Upper row:* Low FSscore (low synthetic feasibility). *Lower row:* High FSscore (high synthetic feasibility).

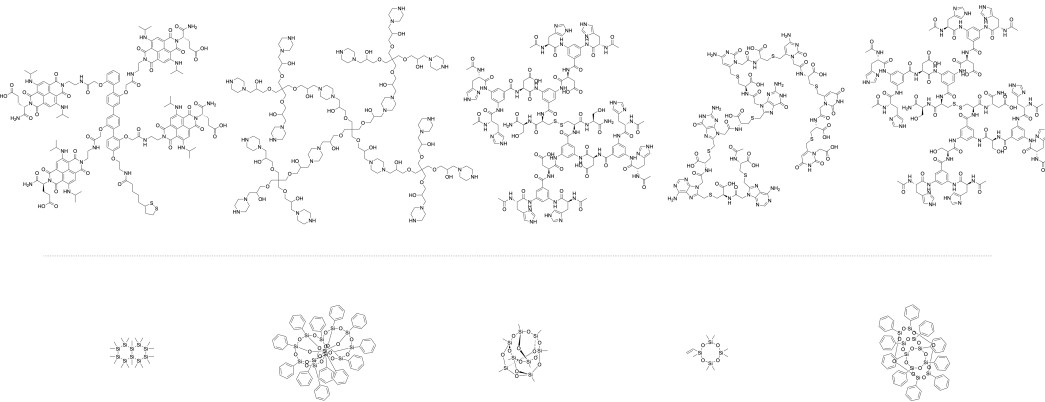

Figure S5.9: Structures from the hold-out test set scored by the pre-trained fingerprint (**Morgan counts**) model. *Upper row:* Low FSscore (low synthetic feasibility). *Lower row:* High FSscore (high synthetic feasibility).

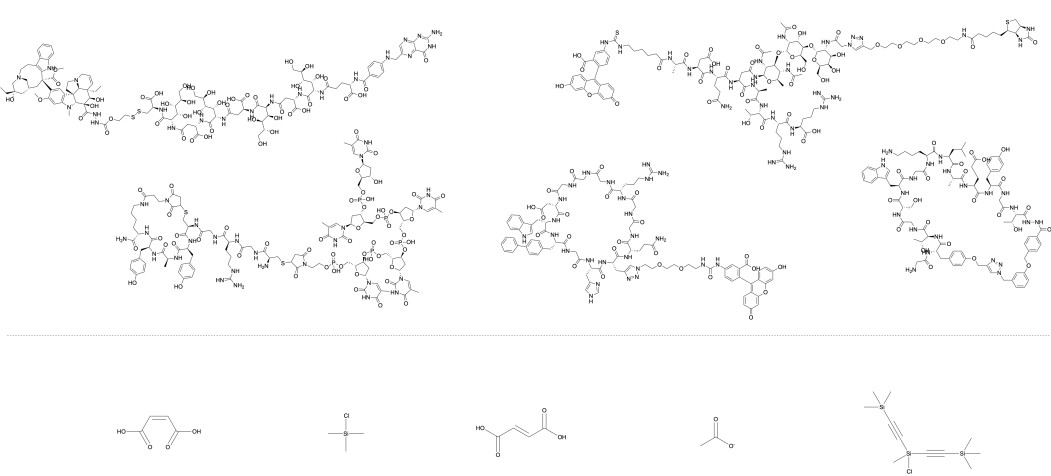

Figure S5.10: Structures from the hold-out test set scored by the pre-trained fingerprint (**Morgan boolean**) model. *Upper row:* Low FSscore (low synthetic feasibility). *Lower row:* High FSscore (high synthetic feasibility).

## S5.2    CP and MC test sets

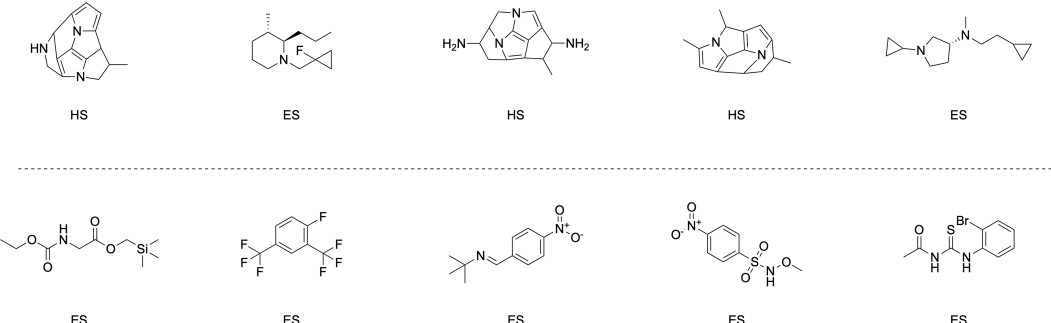

Figure S5.11: Structures from the CP test set scored by the **pre-trained** graph (**GGLGGL**) model. *Upper row:* Low FSscore (low synthetic feasibility). *Lower row:* High FSscore (high synthetic feasibility). The labels below indicate the ground truth.

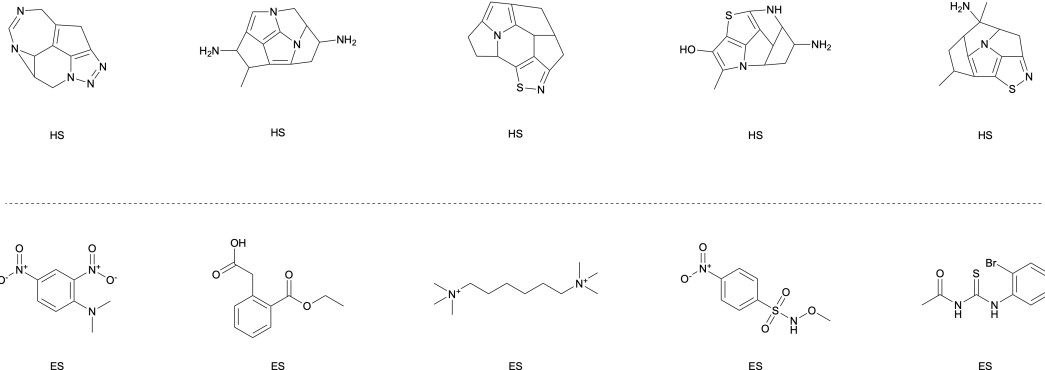

Figure S5.12: Structures from the CP test set scored by the **pre-trained** fingerprint (**Morgan counts**) model. *Upper row:* Low FSscore (low synthetic feasibility). *Lower row:* High FSscore (high synthetic feasibility). The labels below indicate the ground truth.

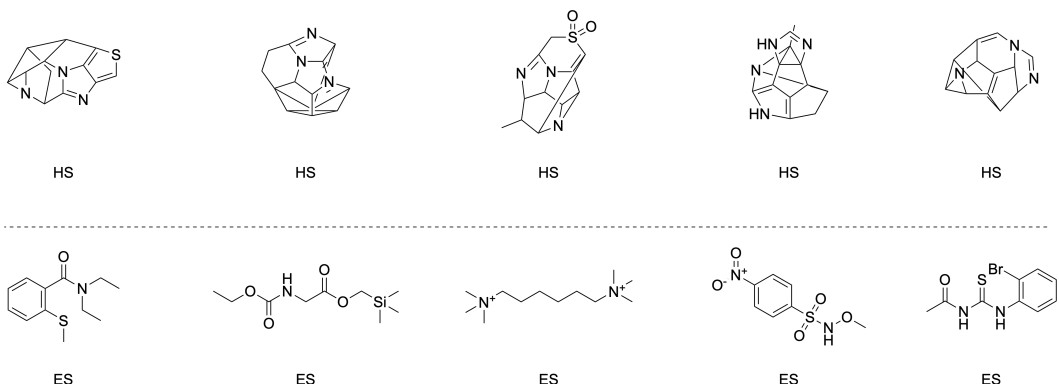

Figure S5.13: Structures from the CP test set scored by the **fine-tuned** graph (**GGLGGL**) model using 50 pairs. *Upper row:* Low FSscore (low synthetic feasibility). *Lower row:* High FSscore (high synthetic feasibility). The labels below indicate the ground truth.

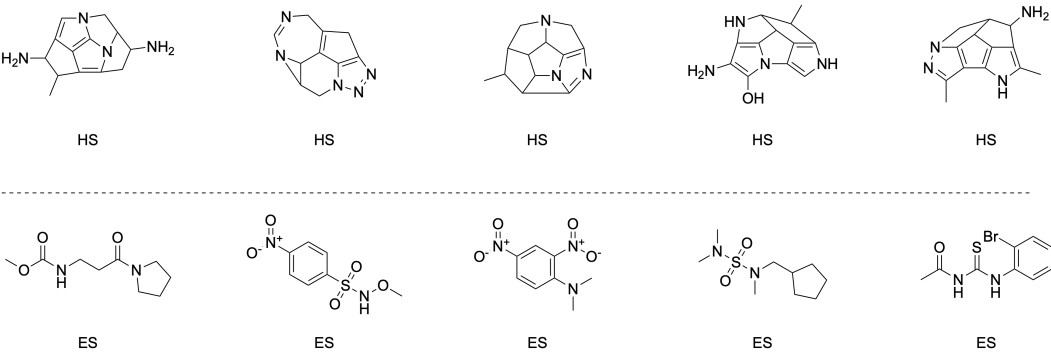

Figure S5.14: Structures from the CP test set scored by the **fine-tuned** fingerprint (**Morgan counts**) model using 50 pairs. *Upper row:* Low FSscore (low synthetic feasibility). *Lower row:* High FSscore (high synthetic feasibility). The labels below indicate the ground truth.

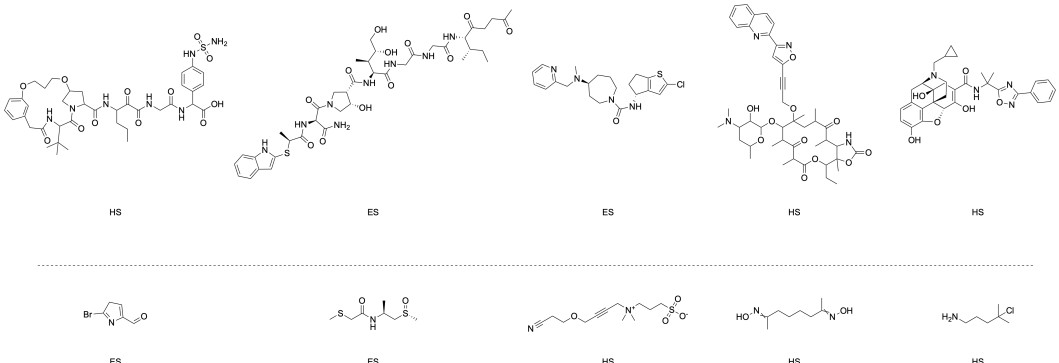

Figure S5.15: Structures from the MC test set scored by the **pre-trained** graph (**GGLGGL**) model. *Upper row:* Low FSscore (low synthetic feasibility). *Lower row:* High FSscore (high synthetic feasibility). The labels below indicate the ground truth.

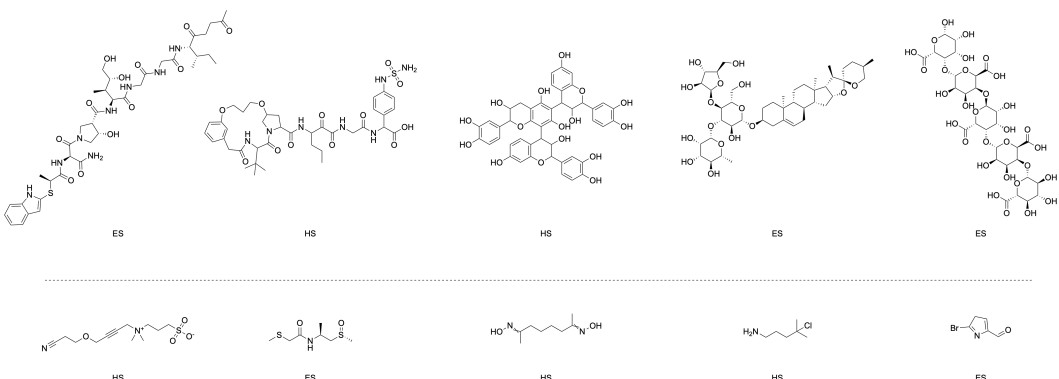

Figure S5.16: Structures from the MC test set scored by the **pre-trained** fingerprint (**Morgan counts**) model. *Upper row:* Low FSscore (low synthetic feasibility). *Lower row:* High FSscore (high synthetic feasibility). The labels below indicate the ground truth.

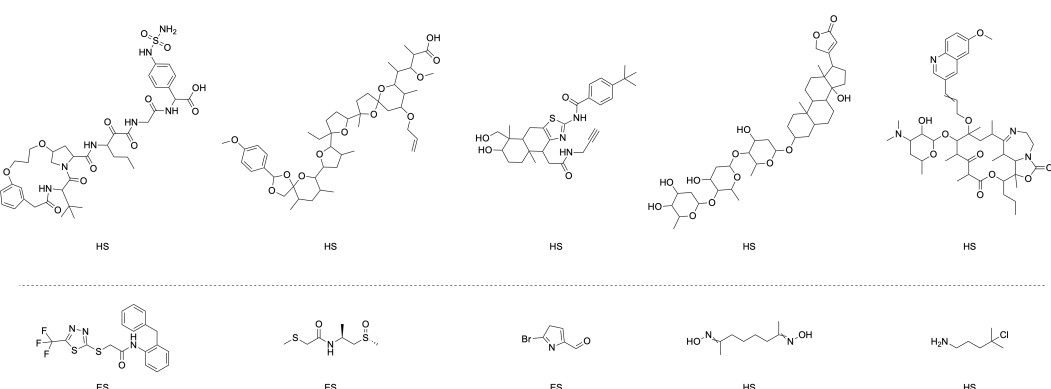

Figure S5.17: Structures from the MC test set scored by the **fine-tuned** graph (**GGLGGL**) model using 30 pairs. *Upper row:* Low FSscore (low synthetic feasibility). *Lower row:* High FSscore (high synthetic feasibility). The labels below indicate the ground truth.

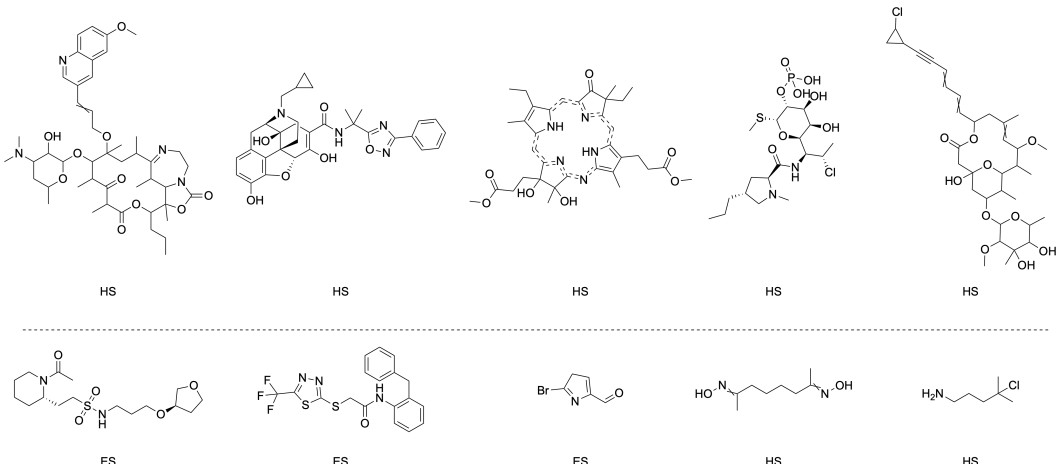

Figure S5.18: Structures from the MC test set scored by the **fine-tuned** fingerprint (**Morgan counts**) model using 30 pairs. *Upper row:* Low FSscore (low synthetic feasibility). *Lower row:* High FSscore (high synthetic feasibility). The labels below indicate the ground truth.

## S5.3 PROTAC-DB

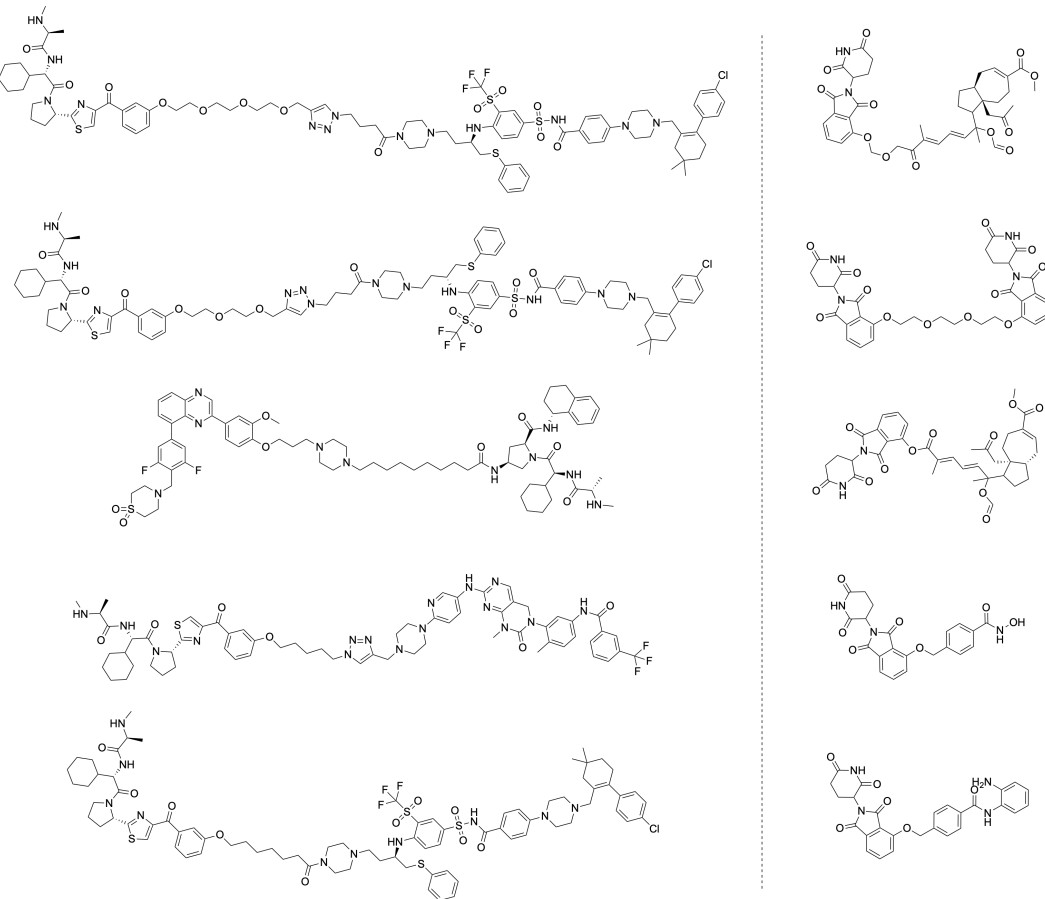

Figure S5.19: Structures from PROTAC-DB scored by the **pre-trained** graph (**GGLGGL**) model. *Upper row:* Low FSscore (low synthetic feasibility). *Lower row:* High FSscore (high synthetic feasibility).

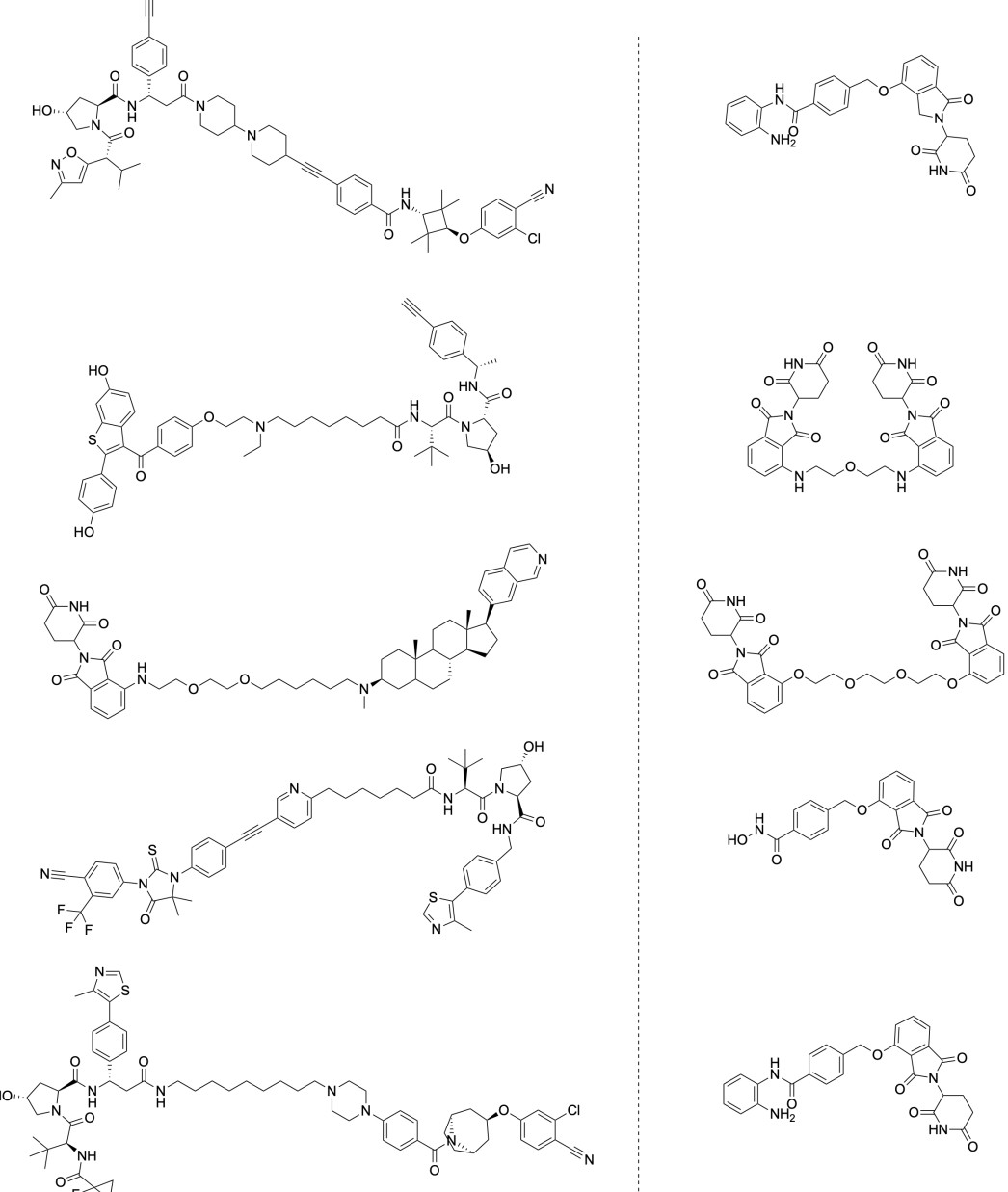

Figure S5.20: Structures from PROTAC-DB scored by the **fine-tuned** graph (**GGLGGL**) model using 50 pairs. *Upper row:* Low FSscore (low synthetic feasibility). *Lower row:* High FSscore (high synthetic feasibility).

## S5.4 REINVENT case study

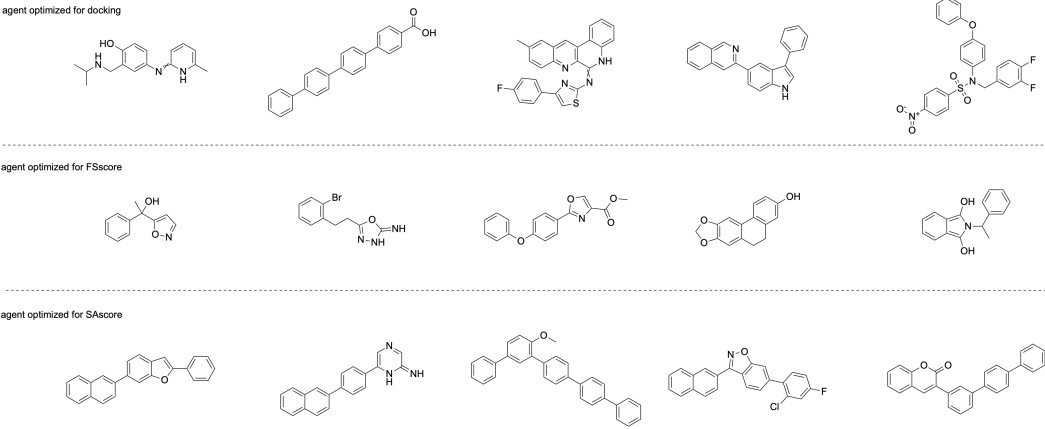

Figure S5.21: Structures from the three agents trained in the REINVENT case study. *Upper row:* Samples from the first agent optimized for docking to DRD2. *Middle row:* Samples from agent optimized to the FSscore.*Lower row:* Samples from agent optimized for SA score.

# S6 Learning curves

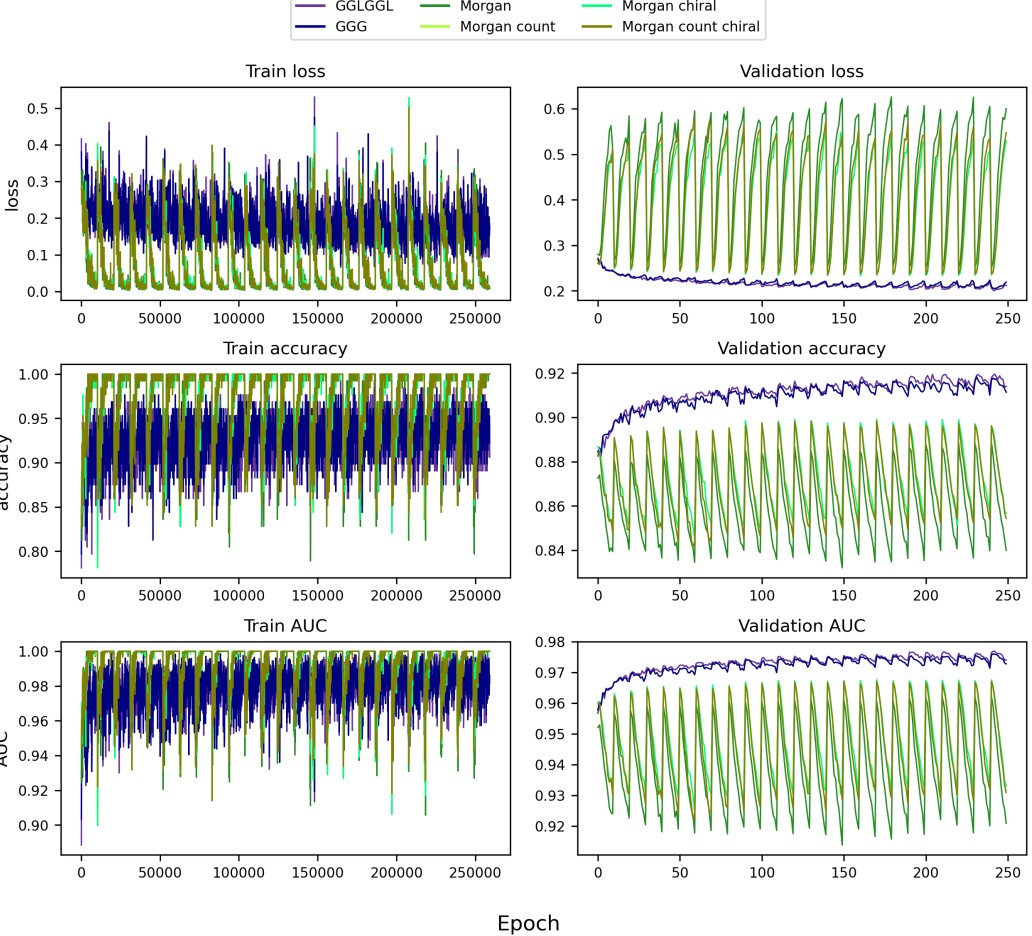

Figure S6.22: Learning curves of the training and validation set for all model variations during **pre-training** for 250 epochs. The training set was split into 25 subsets and the model was trained on each of them for 10 epochs explaining the oscillating nature of the curves.

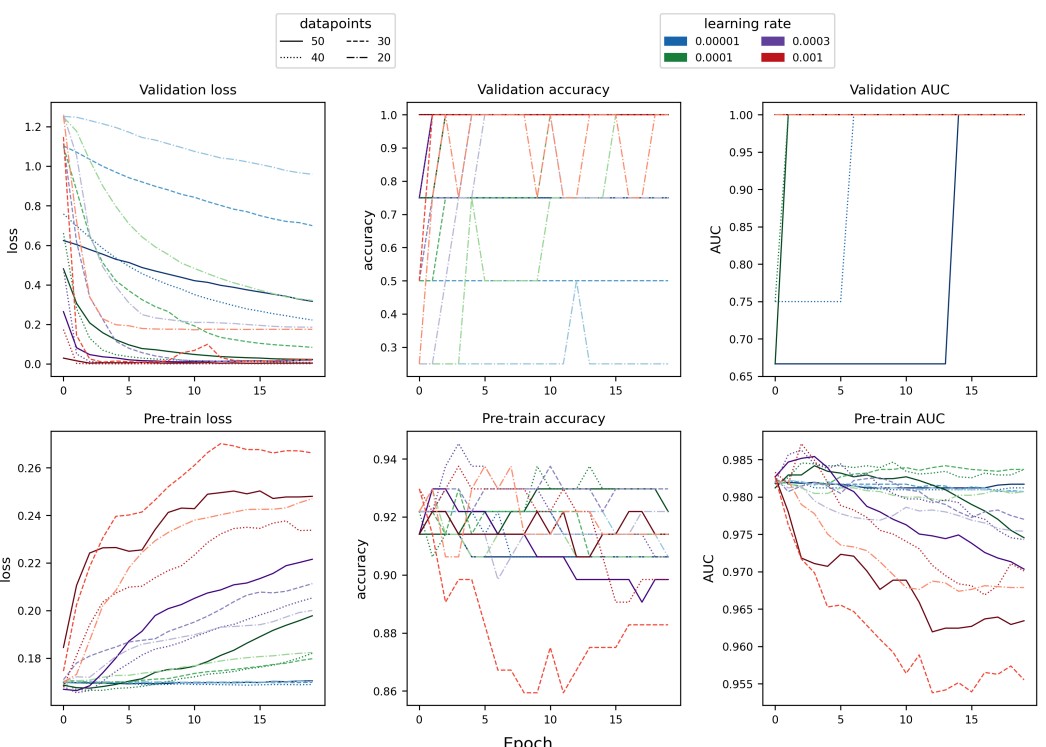

Figure S6.23: Learning curves of fine-tuning the graph-based (GGLGGL) model with varying learning rate and number of pairs used from the **chirality** test set. *Upper row:* Training metrics. *Middle row:* Validation metrics. *Lower row:* Metrics on 5,000 samples from the pre-training test set.

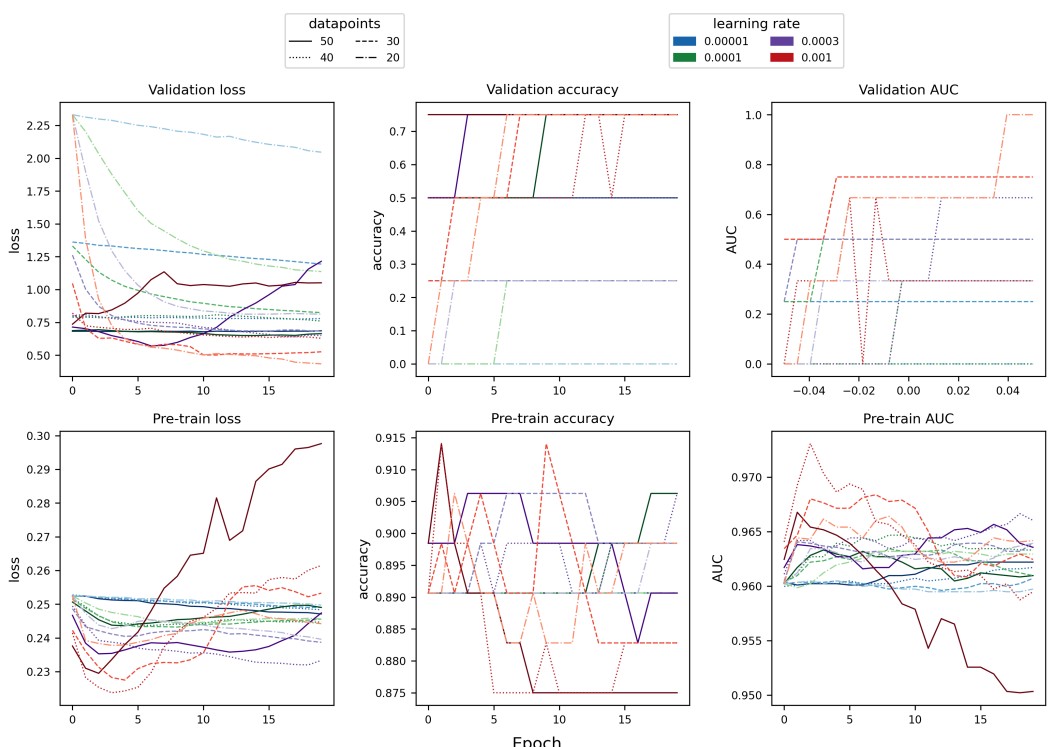

Figure S6.24: Learning curves of fine-tuning the fp-based (Morgan chiral count) model with varying learning rate and number of pairs used from the **chirality** test set. *Upper row:* Training metrics. *Middle row:* Validation metrics. *Lower row:* Metrics on 5,000 samples from the pre-training test set.

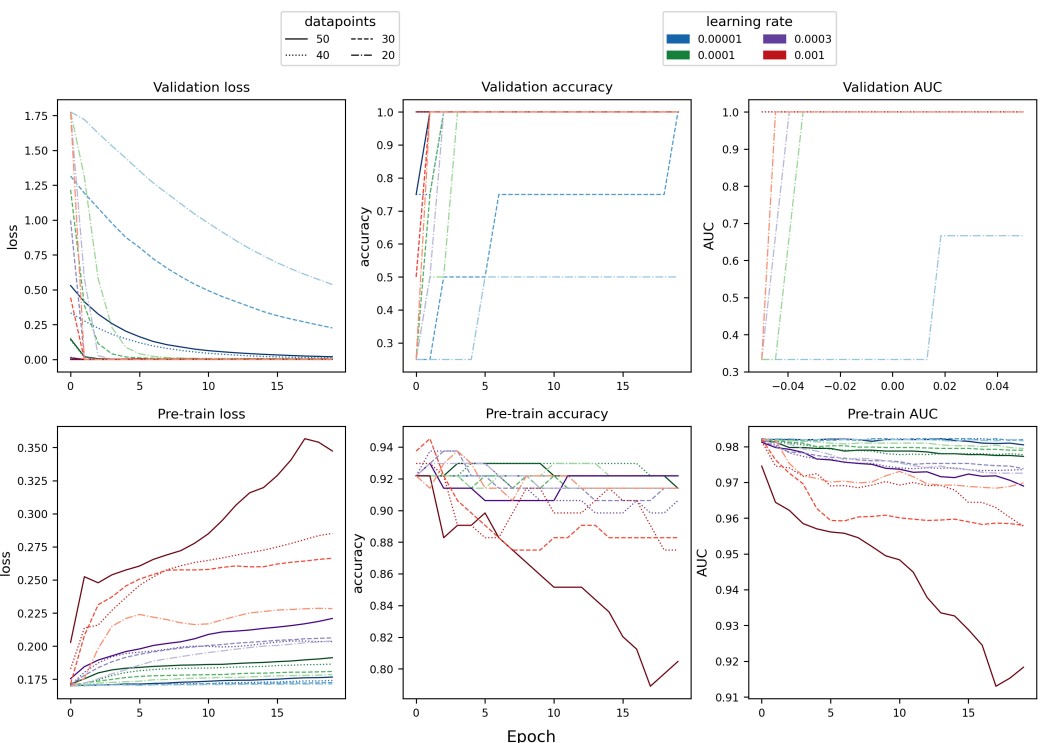

Figure S6.25: Learning curves of fine-tuning the graph-based (GGLGGL) model with varying learning rate and number of pairs used from the **CP** test set. *Upper row:* Training metrics. *Middle row:* Validation metrics. *Lower row:* Metrics on 5,000 samples from the pre-training test set.

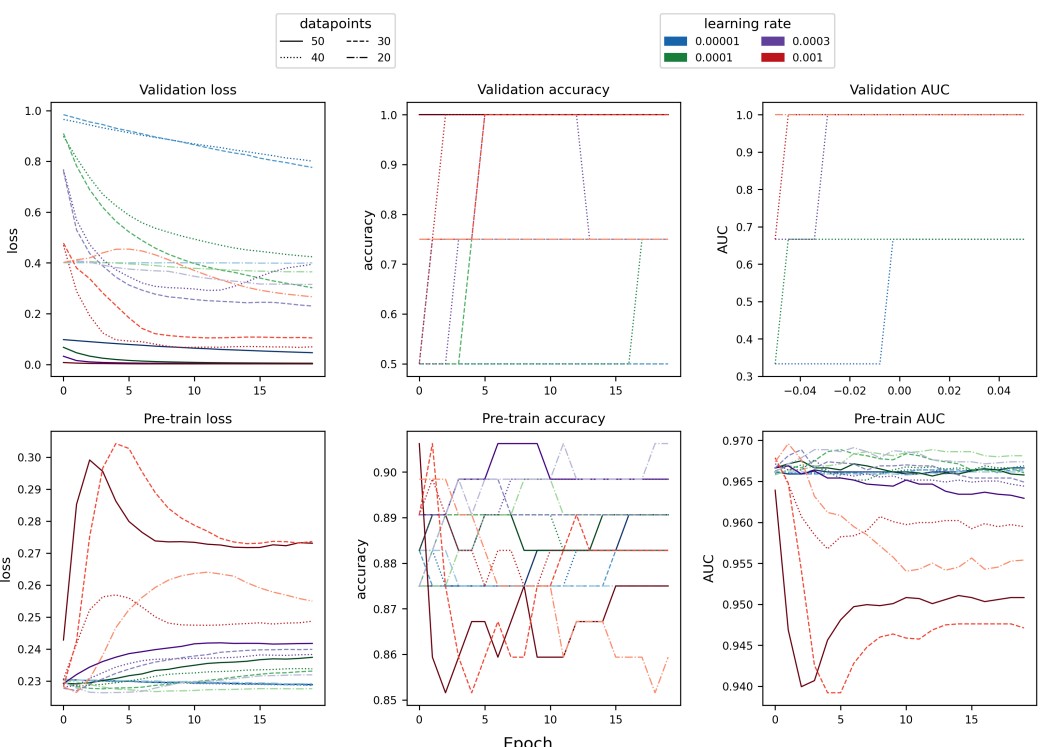

Figure S6.26: Learning curves of fine-tuning the fp-based (Morgan count) model with varying learning rate and number of pairs used from the **CP** test set. *Upper row:* Training metrics. *Middle row:* Validation metrics. *Lower row:* Metrics on 5,000 samples from the pre-training test set.

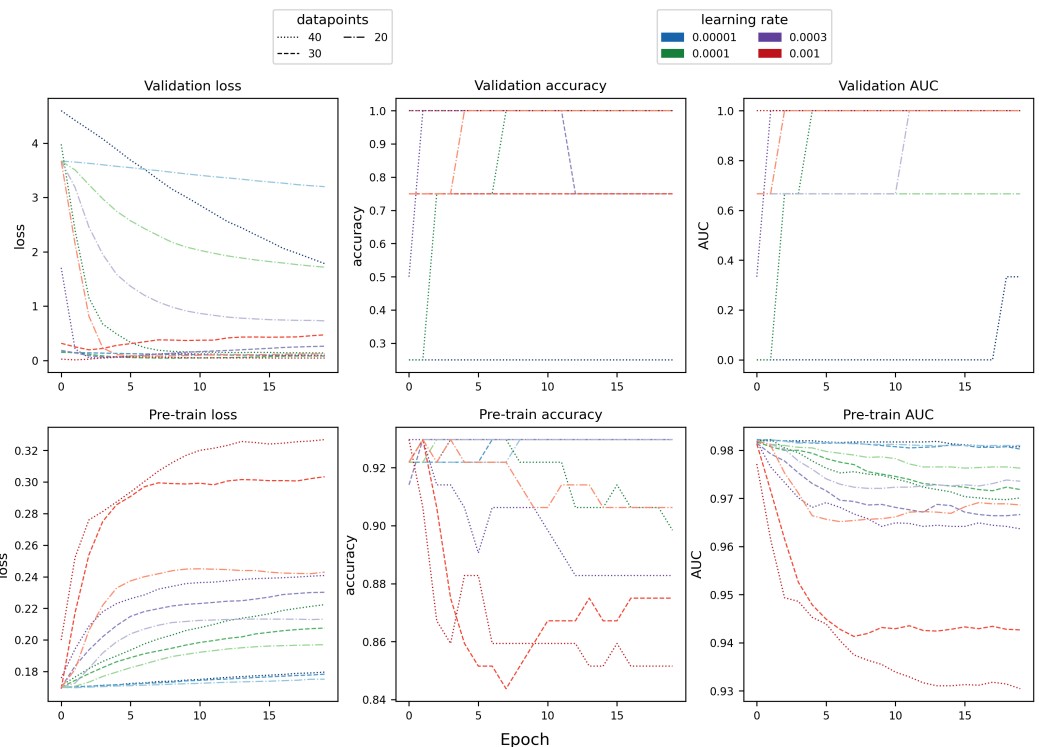

Figure S6.27: Learning curves of fine-tuning the graph-based (GGLGGL) model with varying learning rate and number of pairs used from the **MC** test set. *Upper row:* Training metrics. *Middle row:* Validation metrics. *Lower row:* Metrics on 5,000 samples from the pre-training test set.

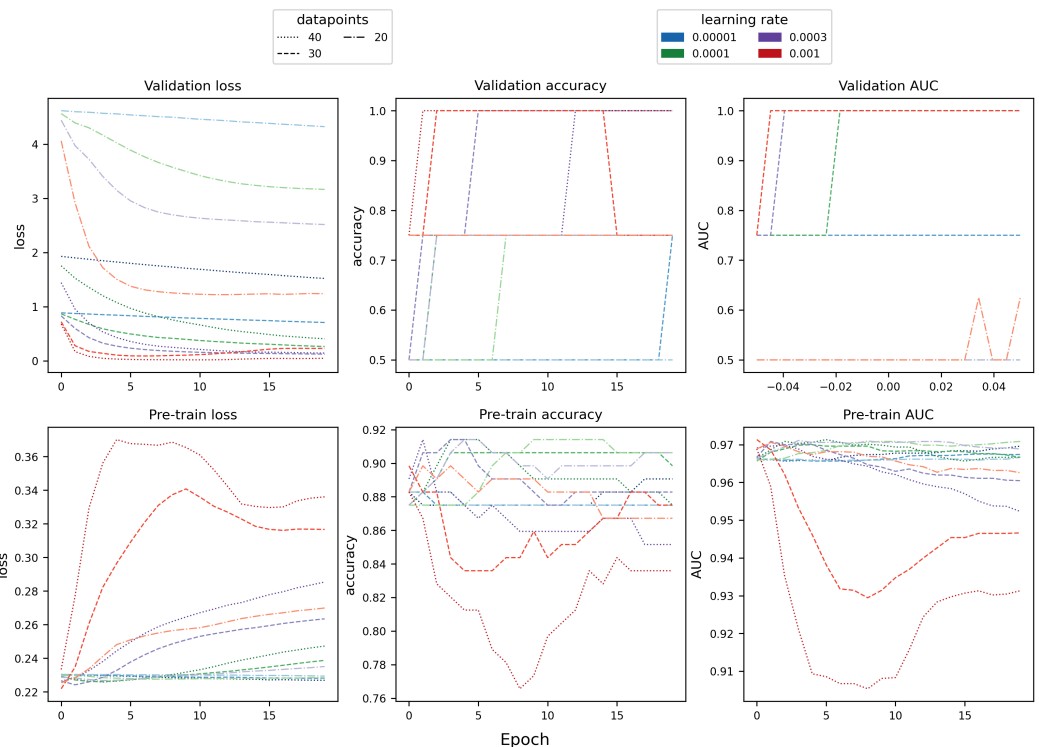

Figure S6.28: Learning curves of fine-tuning the fp-based (Morgan count) model with varying learning rate and number of pairs used from the **MC** test set. *Upper row:* Training metrics. *Middle row:* Validation metrics. *Lower row:* Metrics on 5,000 samples from the pre-training test set.

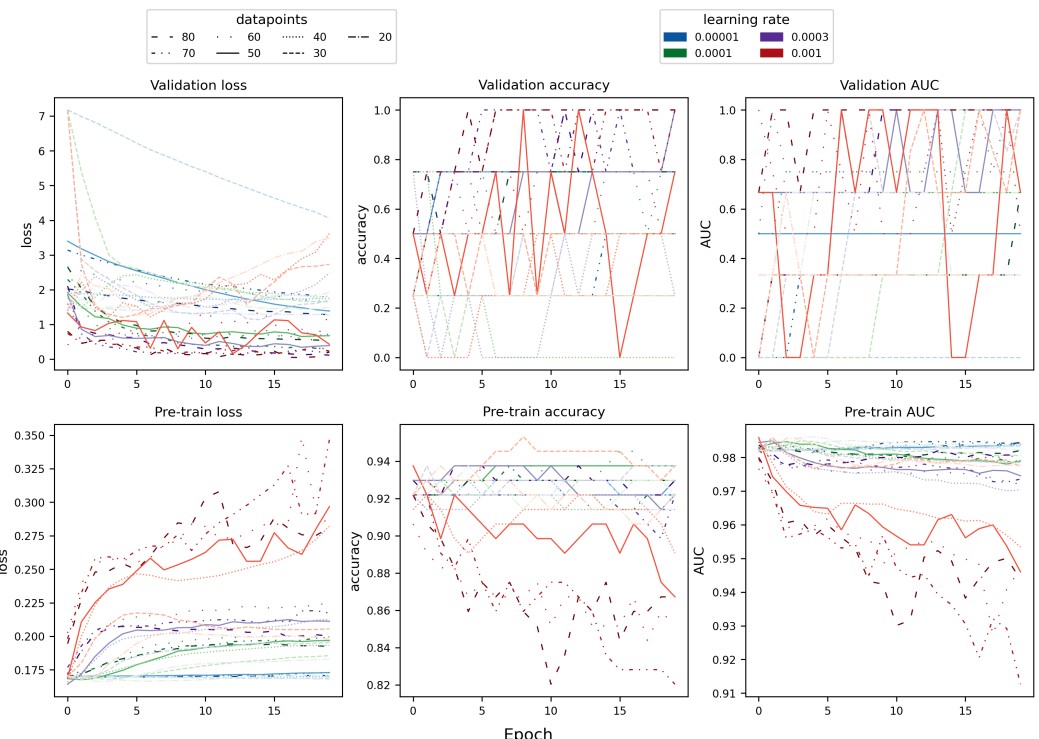

Figure S6.29: Learning curves of fine-tuning the graph-based (GGLGGL) model with varying learning rate and number of pairs used from the **PROTAC-DB** sample. *Upper row:* Training metrics. *Middle row:* Validation metrics. *Lower row:* Metrics on 5,000 samples from the pre-training test set.

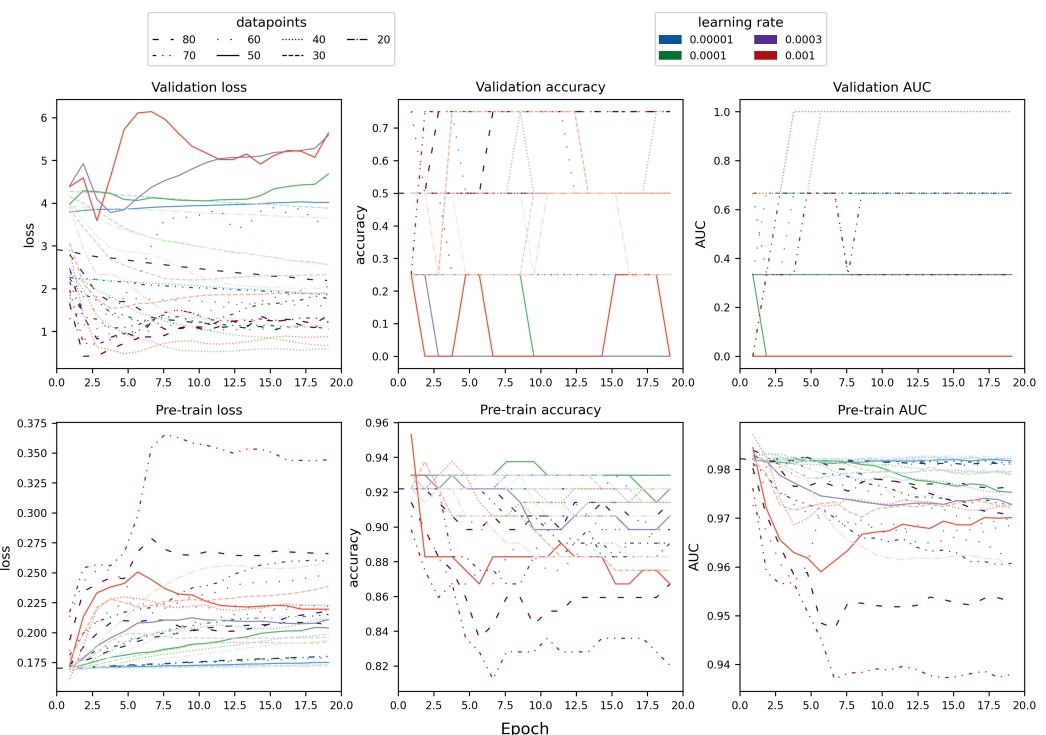

Figure S6.30: Learning curves of fine-tuning the graph-based (GGLGGL) model with varying learning rate and number of pairs used from the **REINVENT** case study. *Upper row:* Training metrics. *Middle row:* Validation metrics. *Lower row:* Metrics on 5,000 samples from the pre-training test set.

