# OpenReview forum: "FSscore: A Machine Learning-based Synthetic Feasibility Score Leveraging Human Expertise"
_NeurIPS.cc/2023/Workshop/AI4Science — NeurIPS2023-AI4Science Poster_

### Meta-Review · Area_Chair_BhtP · 2023-10-27

**Recommendation:** Accept (Poster)
**Confidence:** 3

**Metareview:**

Synthesis is an important task in ML for drug discovery, and this paper proposed a new synthetic complexity score, which is augmented by the human expert. The idea is interesting, though the backbone representation functions are comparatively simple. It would be interesting to see how to extend this to more advanced deep learning models on molecules in the future.